# Precise tea leaf disease detection using UAV low-altitude remote sensing and optimized YOLO11 model

**Yaojun Zhang[1], Guiling Wu****[1]\*, Jianbo Shen[2,3]**

1 School of Information Engineering, Xinyang Agriculture and Forestry University, Xinyang, Henan, China,
2 School of Smart Agriculture Engineering, Wenzhou Vocational College of Science and Technology, Wenzhou, Zhejiang, China, 3 Wenzhou Key Laboratory of AI Agents for Agriculture, Wenzhou, Zhejiang, China

\* guiling@xyafu.edu.cn

## Abstract

Tea leaf diseases seriously affect its yield and quality, and consequently there is an urgent need for intelligent detection methods with high precision and edge deployment capabilities. To address low detection accuracy in complex backgrounds, overfitting due to limited data, and redundant parameters for existing methods, this paper proposes an improved lightweight detection model FCHE-YOLO based on the YOLO11, which aims to achieve rapid and accurate identification of tea leaf disease combining low altitude remote sensing with unmanned aerial vehicle (UAV). The model has made three key optimizations in the structure: Introduce the self-developed lightweight backbone module FC_C3K2, which significantly reduces computation and parameter count while enhancing the robustness of the model to complex scenarios; construct an efficient feature fusion structure HSFPN, optimizing multi-scale information integration and compressing model volume; design the detection head Efficient Head, integrating group convolution and lightweight attention mechanism to improve detection accuracy and suppress overfitting. The experimental results from the self built tea gardens show that the FCHE-YOLO improves the average accuracy (mAP) from 94.1% to 98.1% compared to the benchmark model YOLO11, with an improvement of 4.0 percentage points. Meanwhile, the inference speed of the model increases from 43.3 FPS to 47.5 FPS, with an increase of 9.0%, meeting the real-time detection requirements. More importantly, by network structure optimization, the model's computational complexity is significantly reduced: The floating-point operations per second (FLOPs) decreases from 6.4 G to 4.2 G, with a decrease of 34.3%, and the parameter count decreases from 2.59 M to 1.46 M, with the compression rate reaching 38.9%, which makes the model more suitable for deployment on resource-constrained UAV edge devices. The final test show that

**Data availability statement:** All relevant data for this study are publicly available from the figshare repository (https://figshare.com/s/316807b23895bc3ba3ae).

**Funding:** This research was funded by Science and Technology Research Projects of Henan Province (Grant No. 252102111173) and Cangnan County Modern Agricultural Industry Enhancement Project (Grant No. 2024CNYJY08). The funders had no role in the design of the study; in the collection, analyses, or interpretation of data; in the writing of the manuscript; or in the decision to publish the results.

**Competing interests:** The authors declare no conflicts of interest.

the FCHE-YOLO significantly reduces the missed-detection rate, owns better detection accuracy and deployment practicality, and is suitable for real-time monitoring scenarios of tea leaf diseases with UAVs.

## Introduction

Tea tree is an important agricultural and economic crop, cultivated in approximately 32% of countries and regions worldwide, and now the cultivation areas are continuing to expand. Tea leaves and buds are processed into various products, becoming one of the most popular beverages in the world [1]. However, since tea trees grow in warm and humid environments, they are susceptible to numerous diseases, such as tea brown blight leaf rust, white spot, black rot, red spider mite, and tea mosquito bug. These diseases not only reduce tea yields but also severely affect tea quality, causing significant economic losses for tea farmers [2]. Additionally, the wide variety of tea diseases and their difficult prevention further limit the sustainable development of the tea industry. Currently, tea disease diagnosis is performed manually. Since most tea trees grow in rugged hilly areas, sending professionals to tea gardens for diagnosis is both time-consuming and expensive. When farmers distinguish different types of tea leaf diseases depending on their personal experience, they will obtain the highly subjective results with low diagnostic accuracy and substantial labor in identifying diseased leaves [3].

With the rapid rise of low-altitude UAV remote sensing technology, its combination with computer vision provides a new solution for tea disease detection. UAVs have the characteristics of high flexibility and wide coverage [4–6]. When equipped with ultra-high-definition cameras, they can quickly obtain high-resolution image data in tea gardens. Computer vision technology, especially object detection networks, can process these image data rapidly and efficiently, achieving accurate recognition and classification of tea leaf diseases [7]. The combination of computer vision and UAVs not only performs the real-time monitoring of tea garden disease but also enhances the efficiency and accuracy of detection, providing technical support for precise detection of tea disease [8,9]. This method will effectively reduce the losses caused by diseases, improve the quality of tea, and promote the sustainable development of the tea industry.

In early research, significant progress has been made with computer vision methods to identify leaf diseases. For example, Chaudhary et al. [10] proposed an improved random forest classifier method for multi-class classification of leaf diseases. This method combines the advantages of random forest machine learning algorithm, attribute evaluator method, and instance filtering method, which can effectively detect diseases. Barbudo et al. [11] proposed a plant disease recognition method based on color transformation, color histograms, and classification system. However, tests found that the accuracy rate of this method fluctuates between 40% and 80% in identifying various plant diseases. Hossain et al. [12] developed an image processing system with the Support Vector Machine (SVM) classifiers to recognize the disease. This system is capable of identifying and classifying brown spot disease and algal leaf disease from healthy leaves. S. Prabu et al. [13] adopted the SVM

method, a traditional machine learning algorithm, to detect tea diseases, and utilized the watershed transformation algorithm to segment color-transformed images clearly and applied the gradient feature values of tea images to multi-class support vector machine classifiers for categorizing tea diseases. And the performance was evaluated, proving the effectiveness and good performance of the model. Although these methods perform well in classification accuracy, the acquisition of feature parameters is complex, and certain technical barriers exist. In addition, these traditional detection methods depend on complex feature parameters manually designed or high-performance hardware support, limiting their potential when applied in large-scale agriculture. In order to further reduce application barriers and improve detection efficiency and accuracy, recently the research based on deep learning object detection networks has gradually emerged in the field of crop disease, becoming an important developmental direction.

Object detection networks possess end-to-end learning capabilities and can automatically extract features from data without manual intervention, making the feature extraction process greatly simplified. Currently the mainstream object detection networks can be divided into two categories: single-stage networks (SSD [14], YOLO [15] series (You Only Look Once)) and two-stage algorithms (RCNN [16], Fast RCNN [17], Faster R-CNN [18]). Liu et al. [19] proposed a model called TAME-Faster R-CNN, which combines a trainable attention mechanism explanation (TAME) module with the backbone network of the Faster R-CNN framework, effectively solving the impact of lighting changes on tea disease detection. This network can detect three types of tea diseases (anthracnose, brown spot, and white spot disease) with high detection accuracy, but has a limitation of high computational resource requirement. Gong et al. [20] proposed a region-based Faster R-CNN method to address limitations such as complex background environments, dense features, and small size characteristics of leaf diseases. This method incorporates advanced Res2Net and Feature Pyramid Network architectures as feature extraction networks, and improves the leaf disease detection accuracy by extracting reliable multi-dimensional features. However, this method still has defects such as large model size and deployment difficulties. Two-stage models have a higher accuracy than single-stage models, but they process candidate region generation and feature extraction in two separate steps, resulting in large computational overhead and slow detection speeds, which makes them difficult to deploy on UAVs. In contrast, while single-stage networks lose some accuracy, their higher efficiency and real-time performance have gradually obtained an advantage in object detection, making them more suitable for deployment on edge devices like UAVs for efficient agricultural disease detection in practical environments.

Sun et al. [21] proposed a lightweight MEAN-SSD model that could be deployed on mobile devices for real-time detection of leaf diseases. This model improved detection speed and reduced the number of model parameters by introducing MEAN blocks and Apple-Inception modules. Results showed that MEAN-SSD could achieve 83.12% mAP and 12.53 FPS detection speed, but its performance was limited when processing extremely complex backgrounds and small targets, leaving room for further optimization. Xue et al. [22] proposed an improved model based on YOLOv5 called YOLO-Tea, which incorporated ACMix attention mechanisms, RFB modules, and GCNet networks to improve detection accuracy and speed. However, with a model size of 15.6M, it presented certain difficulties for embedding in edge devices like UAVs. Jan et al. [23] trained the fastest single-stage object detection model, YOLOv7, on a dataset of diseased tea leaves collected from four renowned tea gardens in Bangladesh. Their model could quickly detect tea leaf diseases, but it was prone to false detections when backgrounds were complex. Wang et al. [24] proposed a new lightweight model called YOLOv8-RCAA, which enhanced feature extraction capabilities and inference efficiency by incorporating RepVGG modules, CBAM attention mechanisms, and ATSS modules into the YOLOv8 model. This model ultimately had near real-time inference speed, but its robustness needed to be further improved in extreme scenarios. Gui et al. [25] proposed an improved tea bud detection model in YOLO-GSG based on YOLOX, which incorporated the global context mechanism in the feature fusion network for tea bud discovery in complex backgrounds; the Separate Enhancement Attention Module (SEAM) was added in the detection head to collect information that is beneficial for tea bud classification while suppressing irrelevant information; and the SIOU method was introduced in the post-processing stage to improve the missed detection caused by occlusion. This model achieved a high accuracy rate in detecting tea bud diseases, but its detection efficiency was slow and the model was large, making it difficult to be easily deployed on UAVs. Fang et al. [26] designed

a lightweight tea bud detection model TBF-YOLOv8n to address the issue of high computational complexity in deep learning detection models, which achieved the reduction of model complexity by incorporating DSConv convolution. While the TBF-YOLOv8n model performed excellently in reducing computational load, its real-time performance when processing large amounts of data still needed further optimization.

In summary, current tea disease detection methods face several main challenges: First, the poor detection performance in complex backgrounds and for small targets, with model robustness needing improvement; second, the complex model computations making deployment on UAVs difficult; and finally, the tendency to overfit on small datasets or in complex scenarios. To address these issues, this paper makes relevant improvements based on the YOLO11 model. The main contributions of this research are summarized as follows:

(1) In the backbone network, we incorporate a novel lightweight module FC_C3K2 proposed in this paper. This module combines the FasterNet Block, Convolutional Gated Linear Unit (CGLU), and C3K2 structure. With it, the model's parameter amount and computational load are both reduced, and the robustness is also remarkably improved when handling the inputs under complex backgrounds.

(2) In the neck network, the High-level Screening Feature Pyramid Network (HSFPN) is adopted to replace the traditional feature pyramid structure. HSFPN significantly improves the robustness of the model by effectively integrating high-level and low-level features. And the design of HSFPN further reduces the model size, which makes the improved YOLO11 structure more lightweight and efficient.

(3) In the detection head section, Group Convolution (GConv) is adopted instead of traditional convolution, and an Efficient-Head structure is designed. This structure effectively suppresses the overfitting problem of the model while reducing training parameters and computational complexity, which provides strong support for enhancing detection accuracy and optimizing model performance.

## Materials and methods

Combining UAV remote sensing technology with an object detection model, this paper develops an automated tea leaf disease detection system. The system adopts UAVs to collect image data of tea gardens efficiently, and then employs the improved YOLO11 model to carry out the disease detection and localization, achieving real-time monitoring of tea diseases. The operational process is shown in Fig 1. The core function of the UAV system is to collect high-definition images through real-time aerial photography and ensure comprehensive coverage of all areas in the tea garden. Using the object detection model, image data is rapidly processed and analyzed to identify the types and regions of leaf diseases, facilitating quick and accurate prevention and control of leaf diseases.

To better support disease detection and decision-making, the system incorporates a backend management module. The backend system is capable of recording and displaying the disease category, location, and quantity for each detection area. Managers can view real-time disease data, generate statistical reports, and adjust prevention and control strategies based on the detection results through the backend. In addition, the backend supports flight path planning and management, intelligent pesticide application management, as well as data storage and query functions. It enables long-term tracking of disease occurrence trends and provides data support for the health management of tea plantations.

### Hardware system overview

This paper proposes a tea leaf disease detection system developed based on the integration of an existing agricultural UAV platform, whose hardware composition is shown in Fig 2. The system takes the DJI T25P agricultural UAV as the carrier and integrates a high-performance Jetson TX2 edge computing module, an OAK-D camera, and a GPS module. Among them, the detection system part is self-developed, including modules for image acquisition, disease recognition model inference, data processing, and result uploading; the UAV platform (such as the flight control system, power system, and obstacle avoidance radar) is off-the-shelf equipment.

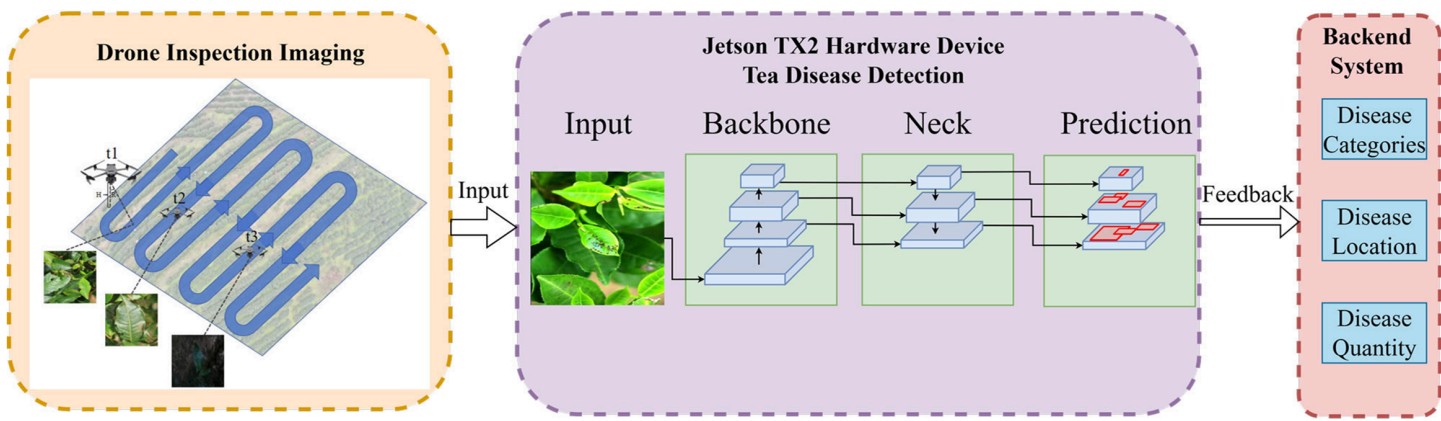

**Fig 1**. Precise detection workflow for tea leaf diseases using UAVs.

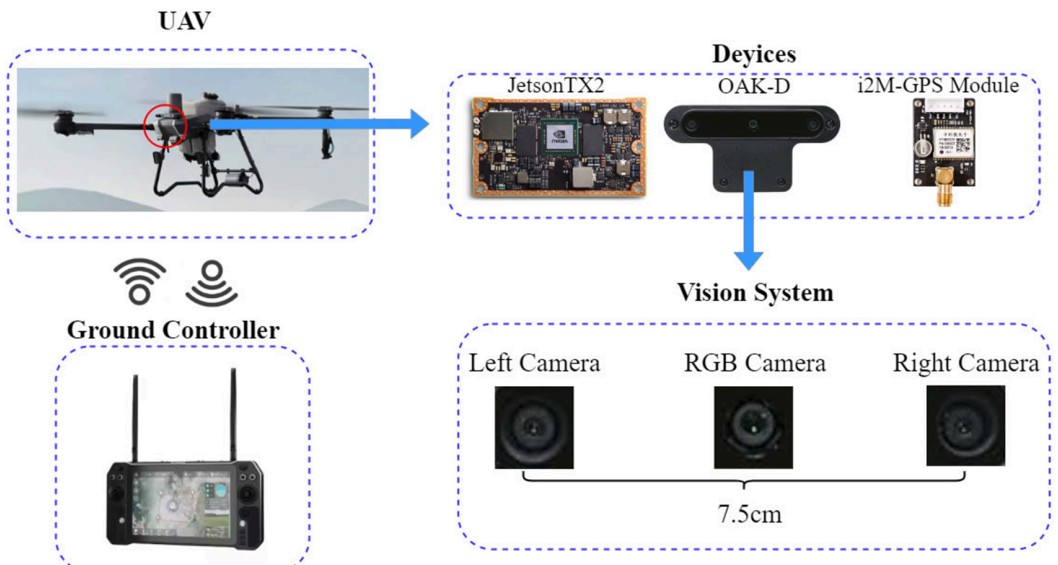

**Fig 2**. Hardware system components.

The OAK-D camera is the core image acquisition device of this system, equipped with a binocular depth camera and an RGB main camera, which can simultaneously output high-definition images and depth map information. This camera supports six-degree-of-freedom spatial perception capability and exhibits good image stability and robustness in the complex environment of tea gardens. Currently, the system adopts a single-camera design to balance the load and performance, but the system architecture supports horizontal expansion to a multi-camera array to meet the operational needs of large areas or multi-angle scenarios.

The Jetson TX2 is the core computing module, featuring high performance and low power consumption, and can complete tasks such as image preprocessing, model inference, and result uploading on the edge side. The GPS module is used for the precise positioning of aerial photography routes. The front and rear obstacle avoidance sensors, downward optical flow sensor, and phased array radar jointly ensure the stability and safety of the UAV in complex environments.

## Tea plantation remote sensing data collection and flight parameter analysis

The experiment was conducted at the Siwangshan Tea Plantation, which has a total planting area of approximately 50,000 square meters. To ensure comprehensive image coverage and sufficient overlap, the flight path was designed using a parallel equidistant route planning method, as shown in Fig 3. The drone flew at intervals of 10 meters, and combined with its visual coverage range, this ensured a high overlap rate and blind-spot-free shooting across the entire planting area, thus obtaining comprehensive high-definition image data.

The so-called "10-meter interval" refers to the lateral distance between adjacent flight routes, which is collectively determined by the field of view (about 120°), flight altitude (10-15 m), and image overlap requirement (> 80%). The UAV flies equidistant along this path, ensuring that each area can be captured at least twice to prevent imaging blind spots and coverage gaps.

In actual data collection, the flight altitude of the drone is set to 10-15 m, which can maintain a stable perspective and image resolution in both flat and sloping environments. This height is the optimal value determined after comprehensively considering the camera resolution (4032 × 3024), crop leaf size, and detection algorithm performance. The flight attitude of the drone is shown in Fig 4, where H is 10 m and R-angle is 60°.

To improve the adaptability of the detection system to changes in external conditions, image acquisition is carried out in different time periods (8:00-11:00, 14:00-17:00) and lighting conditions. The shooting frequency is set to 1 frame per second (1 fps), corresponding to a flight speed of 4-5 m/s approximately. This frequency, which ensures image overlap while reducing redundant images and post-processing burden, is a compromise between image density and system efficiency.

The key parameters for flight and image acquisition used in this study are summarized in Table 1 as follows:

A total of 3,037 original tea leaf images were collected. This quantity was determined comprehensively based on the experimental area (approximately 50,000 m²), UAV flight path planning (10 m spacing, 10–15 m flight altitude), and camera field of view (approximately 120°). Calculated based on the area covered by a single UAV flight and an image overlap rate (>90%), approximately 2,800–3,300 images were theoretically required to achieve full-coverage imaging of the entire tea plantation area and multi-angle redundant sampling. Eventually, 3,037 original images were collected through actual field acquisition.The images have a uniform resolution of 4032 × 3024 pixels and are stored in JPG format. Through a strict quality screening process based on criteria such as image clarity (>0.8), lighting uniformity, and target integrity, 2,432 high-quality disease images were ultimately retained, achieving a screening rate of 80.1%. The distribution of various diseases in the dataset is relatively balanced: 641 images (26.4%) correspond to tea mosquito bug infestation—the largest class—and 142 images (5.8%) correspond to black rot—the smallest class. This distribution helps ensure balance

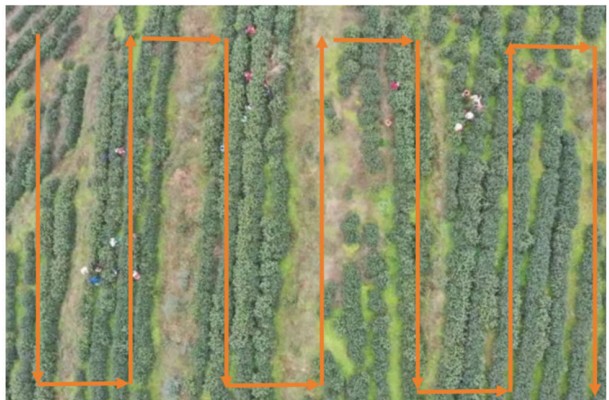

**Fig 3**. UAV flight path.

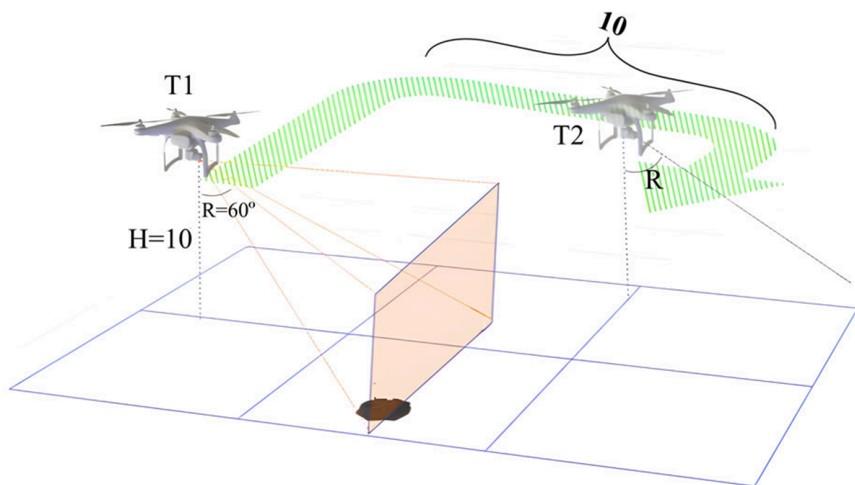

**Fig 4**. **UAV flight attitude during aerial photography.**

**Table 1**. DJI T25P UAV aerial photography parameters.

| Parameter category | Parameter name | Value/range | Specification |
|---|---|---|---|
| Flight parameter | Flight altitude | 10-15 m | Balance field of view coverage and clarity |
| | Flight line spacing | 10 m | Horizontal spacing between adjacent flight routes |
| | Flight speed | 4-5 m/s | Control the overlap density of images |
| | Image collection time period | 8:00-11:00, 14:00-17:00 | Adapt to different lighting conditions |
| Camera parameter | Image resolution | 4032×3024 px | Ensure clear blade characteristics |
| | Shooting frequency | 1 fps | Control data volume and coverage density |
| | Image format | JPG | Balance compression and image quality |
| Sensor | GPS module | High precision positioning | Ensure accuracy of path reproduction and data annotation |
| | Optical flow/obstacle avoidance sensor | Real-time stable flight and safety assurance | Assisted navigation and obstacle avoidance |

and supports the generalization ability of model training.The collected leaf disease images encompass tea black rot, tea brown blight, leaf rust, red spider mite infestation, tea mosquito bug infestation, healthy tea leaves, white spot disease, and other diseases. The labels and corresponding quantities are shown in Table 2, and sample images from the dataset are shown in Fig 5. The images are also collected under different lighting conditions, including strong, normal, and weak light. The different sample images are shown in Fig 6.

**Table 2**. DJI T25P UAV aerial photography parameters.

| Type | Corresponding Label | Number of Images |
|---|---|---|
| Tea black rot | Black rot of tea | 142 |
| Tea brown blight | Brown blight of tea | 95 |
| Tea leaf rust | Leaf rust of tea | 630 |
| Red spider mite infestation | Red Spider infested tea leaf | 307 |
| Tea mosquito bug infestation | Tea Mosquito bug infested leaf | 641 |
| Normal tea leaves | Tea leaf | 207 |
| Tea white spot disease | White spot of tea | 301 |
| Other diseases | disease | 109 |

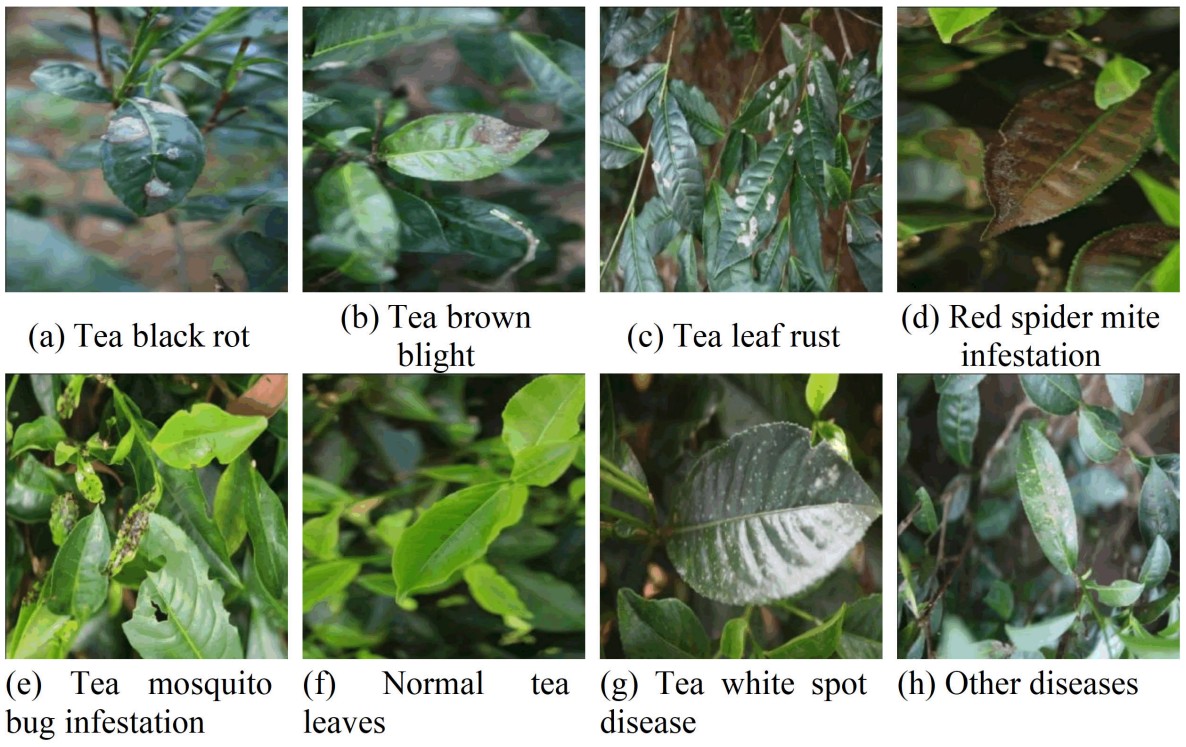

(a) Tea black rot　　　(b) Tea brown blight　　　(c) Tea leaf rust　　　(d) Red spider mite infestation

(e) Tea mosquito bug infestation　　(f) Normal tea leaves　　(g) Tea white spot disease　　(h) Other diseases

**Fig 5**. Sample images of tea leaf diseases.

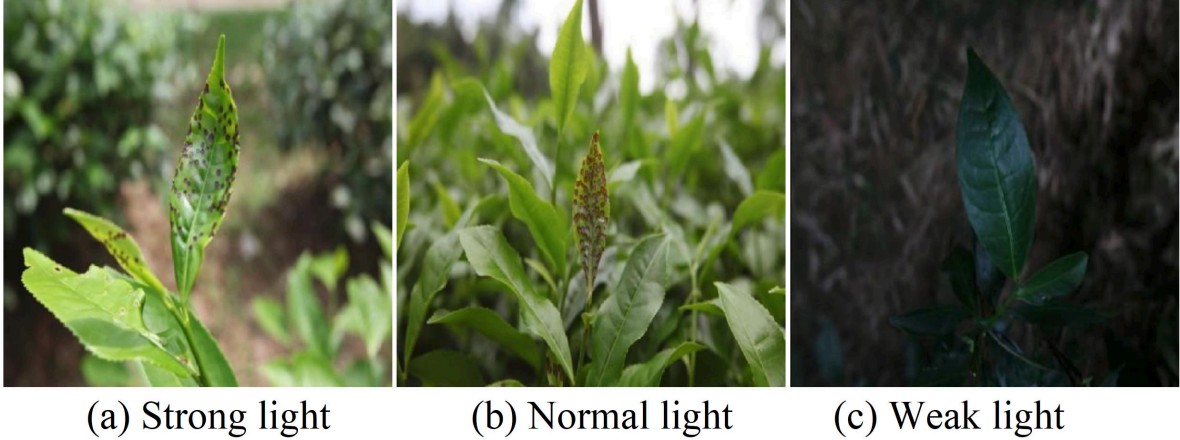

(a) Strong light　　　(b) Normal light　　　(c) Weak light

**Fig 6**. Sample images of different lighting conditions.

## Improved YOLO11 model

### (1) YOLO11 network structure

YOLO11 [27] is the latest version in the YOLO series, optimized from YOLOv8 to achieve better performance on the COCO dataset with fewer parameters. The network structure of YOLO11 is shown in Fig 7, which specifically comprises

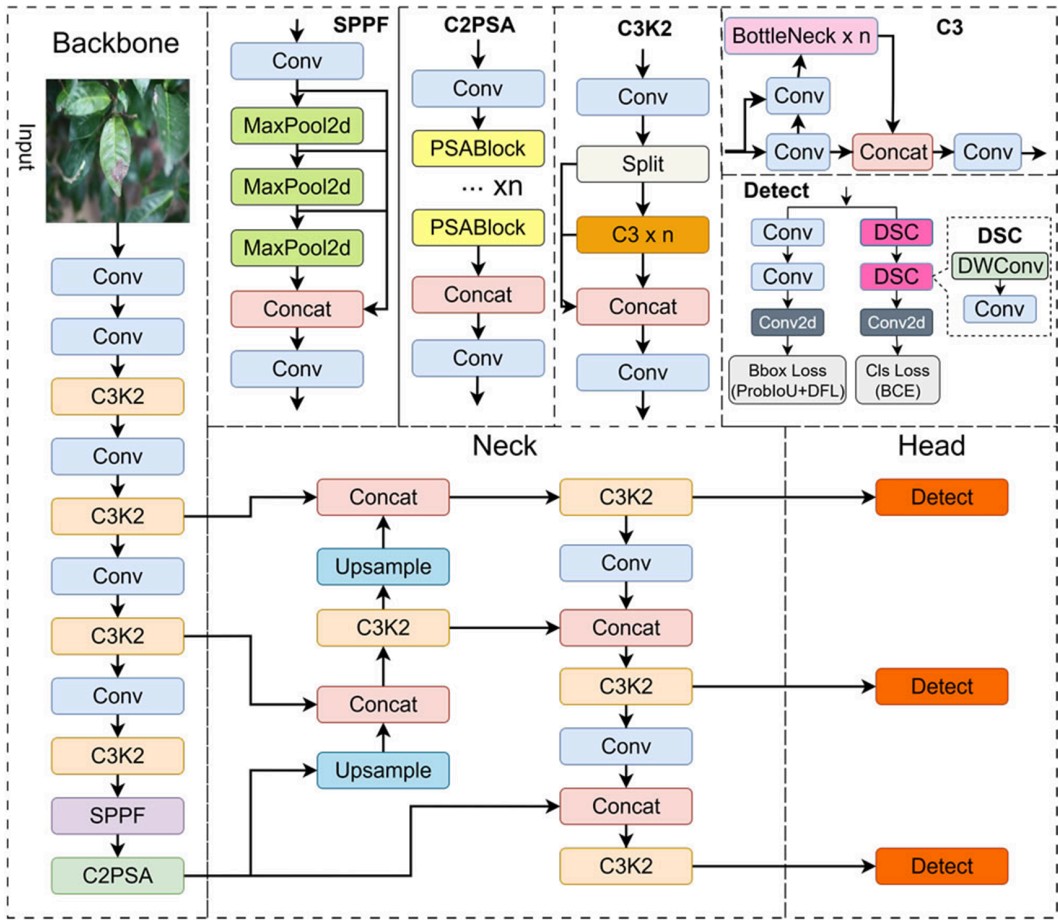

**Fig 7. YOLO11 model structure diagram.**

three core modules: backbone network, feature fusion layer, and detection head. The Backbone is responsible for extracting multi-scale features from input images, the Neck fuses features of different scales to enhance detection performance, and the Detection Head converts the processed feature maps into target bounding boxes and category predictions, enabling the final detection results.

The Backbone is the fundamental component of YOLO11, and its main structure includes the Conv (Convolutional Layer) module, C3K2 module, SPPF (Spatial Pyramid Pooling Fast) module, and C2PSA (Cross-stage Partial Spatial Attention) module, ensuring the model can efficiently extract local and global features. Firstly, the Conv (Convolutional Layer) module is the initial part of the backbone network, responsible for basic low-level feature extraction of input images and providing a foundation for subsequent high-level feature extraction. Next, the C3K2 module integrates multiple convolutional layers and bypass connections, and adopts flexible convolutional configurations and bottleneck designs to improve feature extraction capability. By introducing the Cross Stage Partial (CSP) architecture, the C3K2 optimizes gradient flow, enhances local feature extraction, improves training stability, and strengthens the robustness of YOLO11 in complex backgrounds. Finally, the SPPF (Spatial Pyramid Pooling Fast) module enhances the model's ability to capture global contextual information through multi-scale pooling operations, while the C2PSA (Cross-stage Partial Spatial Attention) module improves the model's detection ability in complex scenes by effectively filtering irrelevant information and emphasizing key features—exhibiting significant advantages especially in multi-target detection and occlusion problems.

The Neck module connects the Backbone and Detection Head, responsible for the fusion of multi-scale features and particularly suitable for detecting objects of different sizes. Its core task is to enhance object detection performance and ensure the model maintains efficient performance across multiple scales. Firstly, the Upsample (Up-sampling) module helps the model better detect smaller objects by increasing the resolution of low-resolution feature maps. Secondly, the Concat (Feature Concatenation) module fuses feature maps with different resolutions, enabling the model to utilize multi-scale information to improve detection accuracy. Finally, the C3K2 module in the Neck enhances feature representation capability and improves detection accuracy by performing repeated convolution operations and shortcut connections for cross-scale feature fusion. The introduction of the C3K2 module allows the model to achieve stronger expression ability across different scales, thereby improving overall detection performance.

The Detection Head is the final stage of YOLO11, responsible for converting processed feature maps into accurate object detection results. With its multi-scale capability, it ensures the model exhibits excellent performance in object detection tasks involving targets of various sizes. Firstly, the classification detection head adopts the Depthwise Separable Convolution (DWConv) structure—a lightweight convolution method that improves computational efficiency by simplifying convolution operations. Through the combination of depthwise convolution and pointwise convolution, DWConv reduces computational cost while maintaining high detection accuracy. Next, the bounding box detection path captures precise spatial information through standard convolution, ensuring accurate prediction of target boundaries. Finally, the classification detection path utilizes DWConv for classification tasks, reducing computational complexity and making it particularly suitable for resource-constrained environments such as edge devices or embedded systems. While ensuring accuracy, DWConv significantly reduces computational cost, enabling YOLO11 to efficiently handle multi-target scenes.

**(2) C3K2-Faster-CGLU structure**

Since tea leaf images captured by UAVs have relatively complex backgrounds, models need stronger robustness and feature extraction capabilities when processing such images. Typically, when dealing with complex background images, network structures are deepened to achieve better results. However, for resource-constrained edge devices like UAVs, this approach is not practical. To address this issue, combining the FasterNet Block, Convolutional Gated Linear Unit (CGLU), and C3K2 structure, this paper proposed a new lightweight structure called C3K2-Faster-CGLU (abbreviated as FC C3K2). This improvement not only effectively reduces the parameter number and size of the model, but also significantly enhances the model's robustness.

**1) FasterNet block structure.** FasterNet [28] is an efficient neural network architecture that employs the innovative Partial Convolution (PConv) [29] technique. PConv reduces computation and memory access by processing only a portion of the input channels, as shown in Fig 8. FasterNet is a four-layer hierarchical architecture, with each layer composed of a series of FasterNet Blocks and supplemented with embedding or merging operation layers. Within each FasterNet Block, the core component is the PConv layer, subsequently enhanced by two regular convolution (Conv) layers.

In the FasterNet architecture, FasterNet Blocks are repeatedly used to enhance the network's feature extraction capability. Each block is responsible for extracting features from different levels, by which the network is capable of handling more complex and dynamically changing inputs without significantly increasing computational cost. Reusing FasterNet Block blocks has several advantages as follows:

(1) Improving robustness: Each FasterNet Block extracts features from different scale and level, which can effectively enhance the model's adaptability to complex background.

(2) Optimizing computing resources: Due to that PConv reduces the computational load and memory access of each convolutional layer, reusing these blocks not only avoids redundant computing, but also improves the inference efficiency of the network on edge devices.

(3) Reduce computational redundancy: By reusing the FasterNet Block, FasterNet can improve the model's ability to capture details and express features without increasing excessive computing resources.

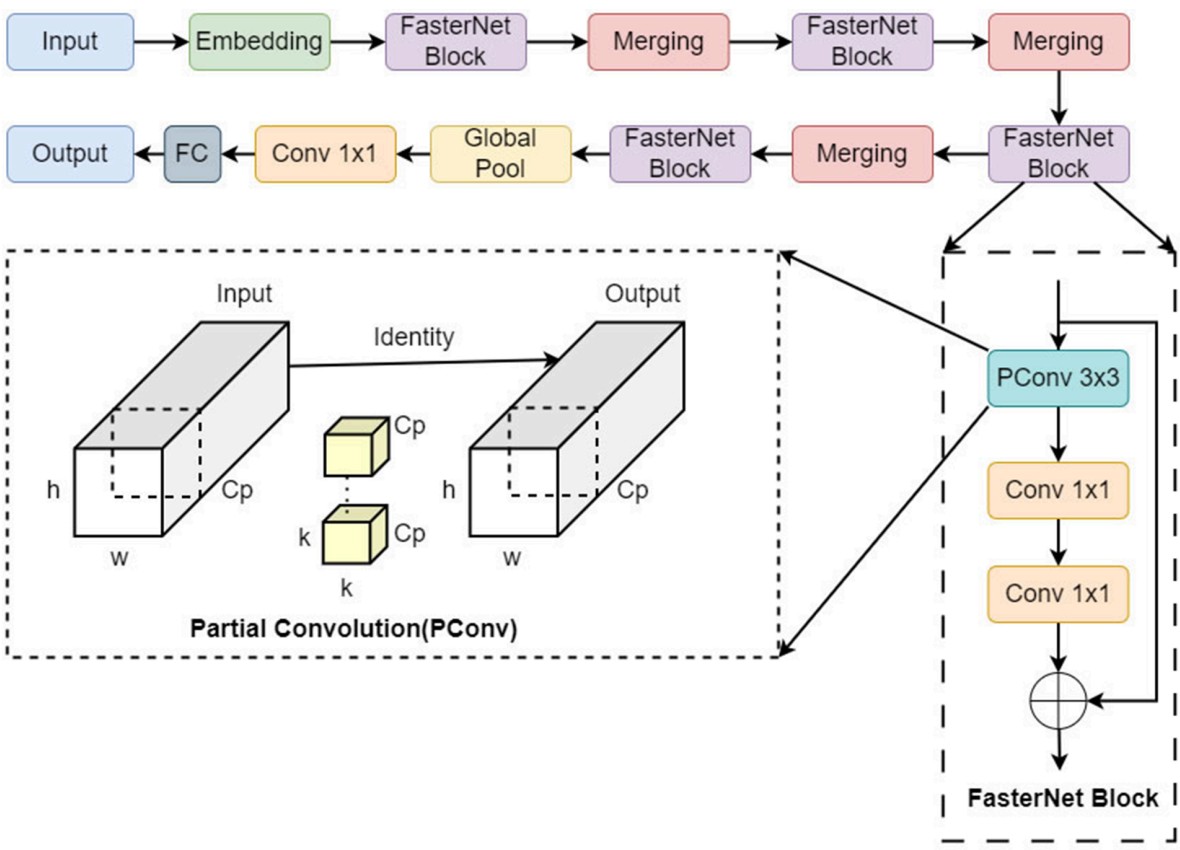

**Fig 8. FasterNet structure diagram.**

Assuming the number of channels for both input and output feature maps is $c$, the convolution kernel size is $k$, and $c_p$ is the number of channels in PConv, the FLOPs (floating point operations) for PConv can be calculated as:

$$FLOPs = h \times w \times k^2 \times c_p^2 \tag{1}$$

When $c_p = \frac{c}{4}$, the FLOPs of PConv are only 1/4 of standard convolution. Furthermore, the memory access (MA) required by PConv is:

$$MA \approx h \times w \times 2c_p + k^2 \times c_p^2 \approx h \times w \times 2c_p \tag{2}$$

From the above calculations, it can be found that the memory access of PConv is only one-fourth that of conventional convolution, significantly decreasing computational and memory overhead. This design enables the FasterNet Block to ensure high efficiency while dramatically reducing computation and memory consumption, making it especially suitable for edge devices and real-time processing tasks.

**2) Convolutional gated linear unit.** The Convolutional GLU (CGLU) [30] fuses channel attention mechanism with local image features, improving the local feature capture ability and robustness of the network. whose structure is shown in Fig 9. It can be seen that the CGLU is composed of two linear projections that are integrated via element-wise intelligent multiplication, creating a gated channel attention mechanism that relies on adjacent features. The distinctive benefit of CGLU is its ability to produce unique gating signals for each token using fine-grained adjacent features, which resolves

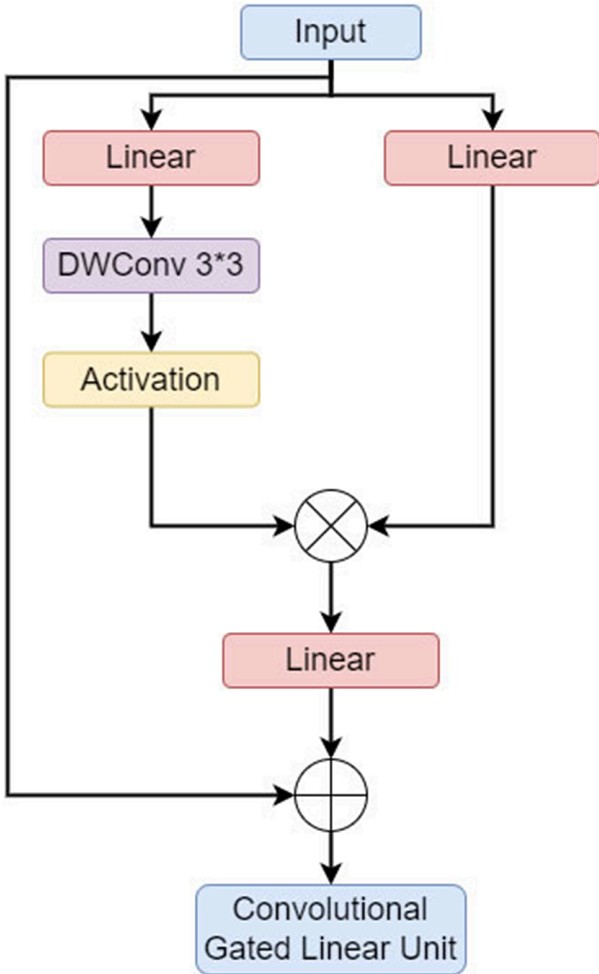

**Fig 9**. CGLU structure diagram.

the coarse-grained problems linked to global average pooling and further improves the model's sensitivity to details and expressiveness.

**3) FC_C3K2 structure.** Combining the FasterNet Block structure with the Convolutional Gated Linear Unit (CGLU), as shown in Fig 10(a), enables the model to adaptively modify the flow of information in response to varying input characteristics. This configuration not only enhances the adaptability of the model in handling complex or variable inputs, but also retains the lightweight characteristics of the structure. Faster-CGLU makes the feature extraction more efficient and accurate, and the overall function of the network greatly enhanced.

The proposed Faster-CGLU module was combined with the C3K2 structure of YOLO11 to construct the C3K2-Faster-CGLU module (abbreviated as FC_C3K2 module), as shown in Fig 10(b). When ensuring lightweight properties, this module adjusts the weights of different input features dynamically, enhancing the adaptability of the model in handling complex or variable inputs. This design not only increases the flexibility of the model, but also ensures its efficient performance in various tasks and environments.

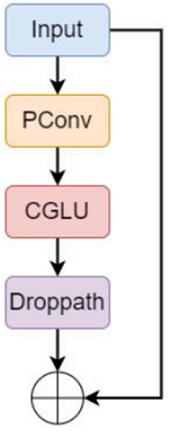
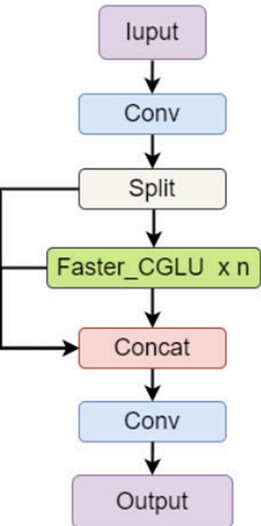

(a) Faster-CGLU structure diagram　　(b) FC_C3K2 structure diagram

**Fig 10**. **Faster-CGLU and FC_C3K2 structure diagrams.**

**(3) High-level Screening Feature Pyramid Network (HSFPN)**

Aiming at the multi-scale problem of tea leaf diseases, this paper puts forward a Hybrid Scale Feature Pyramid Network (HSFPN) which combines advanced screening and feature fusion [31]. This structure effectively screens features maps of various scales, while reducing the number of parameters, synergistically fusing information of different scales, and enhancing the detection capability of the model.

HSFPN is a network architecture proposed based on the improved mechanism MFDS-DETR (Multi-level Feature Fusion and Deformable Self-attention DETR). Its core design comprises two units: a feature selection module and a feature fusion module, which process the tasks of feature extraction and feature fusion respectively. The HSFPN architecture is shown in Fig 11.

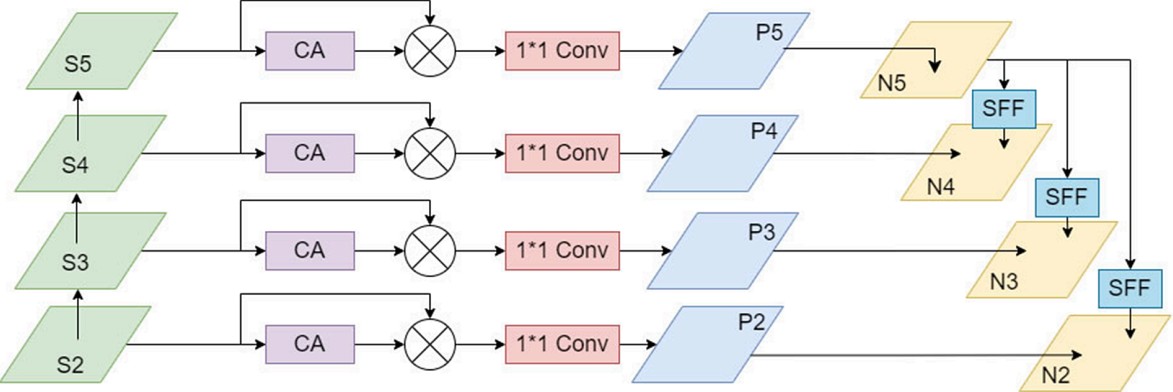

**Fig 11**. **HSFPN structure diagram.**

Firstly, through the feature selection module, the High-level Screening-Feature Pyramid Network (HS-FPN) processes the input feature maps to extract key information related to the target and filter out the irrelevant data. This module utilizes a Channel Attention (CA) module [32] to process the input feature maps, fusing the feature maps produced by global average pooling and global maximum pooling. Then, a Sigmoid activation function is used to calculate weight values for each channel to obtain the final weights (the CA module structure is shown in Fig 12). The CA module uses this method to accurately extract key information about tea leaf diseases from each channel while avoiding further information loss. Subsequently, the weight information is multiplied with the feature maps with corresponding scale, generating the filtered feature maps. Prior to feature fusion, $1 \times 1$ convolutions are used to reduce the number of channels in each scale feature map to 256 so as to guarantee the alignment between feature maps of varying scales. The CA is represented as follows:

$$f_{in} \in R^{C \times H \times W} \tag{3}$$

$$f_{CA} \in R^{C \times 1 \times 1} \tag{4}$$

where $f_{in}$ represents the input feature map, $f_{CA}$ represents the final weights for each channel, and $C, H$, and $W$ represent the channel count, height, and width of the feature map, respectively.

After extracting features from the feature maps, basic semantic information is removed from low-level features employing high-level features as weights, thereby achieving targeted feature fusion. Given high-level features $f_{high}$ and low-level features $f_{low}$, a transposed convolution with stride 2 and kernel size $3 \times 3$ is first applied to the high-level features $f_{high}$ to obtain $f_{high}$. To guarantee dimensional alignment of the features with high and low levels, $f_{\overline{high}}$ undergoes bilinear interpolation upsampling or downsampling to generate aligned features $f_{att}$. Ultimately, the features of low level, filtered by the CA module, are combined with the aligned features of high level $f_{att}$ to generate the output features $f_{out}$. The formula process is as follows:

$$f_{low} \in R^{C \times H_1 \times W_1} \tag{5}$$

$$f_{high} \in R^{C \times H \times W} \tag{6}$$

$$f_{\overline{high}} \in R^{C \times 2H \times 2W} \tag{7}$$

$$f_{att} = \text{Resize}\left(f_{\overline{high}}, \text{ size } = f_{low}\right) \tag{8}$$

$$f_{out} = \text{Fuse}\left(f_{att}, CA\left(f_{low}\right)\right) \tag{9}$$

where Resize(.) represents the bilinear interpolation operation that aligns high-level features with low-level features in spatial dimensions. $CA\left(f_{low}\right)$ represents the processing of low-level features $f_{low}$ through the Channel Attention module, and Fuse(.) represents the feature fusion operation (such as element-wise addition or concatenation).

By constructing a multi-scale fusion network based on feature selection and feature fusion, effective integration of filtered low-level features with high-level features can be achieved, realizing multi-scale feature fusion and significantly enhancing the detection capability of the model. Furthermore, semantic information extracted from high-level feature

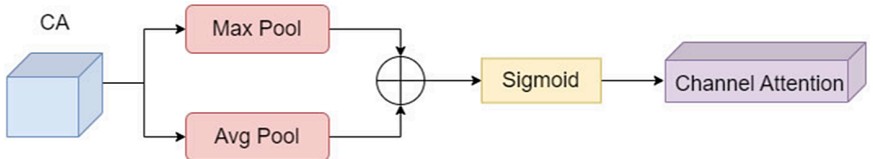

**Fig 12**. **CA module structure diagram.**

maps is adopted as weights to guide the selection and fusion of low-level features, significantly improving the efficiency of feature fusion within the network.

## (4) Efficient-head detection head

The detection head in YOLO11 adopts a decoupled head architecture with two distinct branches responsible for classification and bounding box regression. The bounding box regression consists of two $3 \times 3$ Conv layers and one $1 \times 1$ Conv 2 d layer, while the classification task consists of two $3 \times 3$ DSConv (Depthwise Separable Convolution) modules and one $1 \times 1$ Conv 2 d layer. Compared to coupled heads, this design improves overall detection capability. However, it significantly increases the number of parameters and computational complexity. The detection head in YOLO11 accounts for approximately one-fourth of the total parameters of the model.

Given the computational resource constraints of UAVs, there is an urgent need to introduce high-performance, lightweight convolution structures to replace convolutions in the detection head. To this end, this paper adopts Group Convolution (GConv) [33] to replace traditional convolution, designing an Efficient-Head [34] structure. The Efficient-Head module is shown in Fig 13.

GConv functions similarly to regularization, effectively preventing model overfitting while reducing training parameters and computational complexity. The number of parameters and computational complexity of GConv are approximately $1/g$ of regular convolution, where $g$ represents the number of groups. In GConv, feature maps are divided into $g$ groups, and convolution kernels are similarly divided into $g$ groups. Convolution is performed within respective groups, and outputs produced from each group are combined along the channel dimension, as illustrated in Fig 14. Theoretically, both the parameters and computational complexity of GConv are $1/g$ of regular convolution. The number of parameters for regular convolution is shown in Equation (10), and FLOPs (floating-point operations) in Equation (11). The number of parameters for GConv is shown in Equation (12), and FLOPs in Equation (13). By comparing Equations (10) and (12), GConv's number of parameters is only $1/g$ of regular convolution. Similarly, comparing Equations (11) and (13), its computational complexity is also reduced to $1/g$ of regular convolution. The value of $g$ is a hyperparameter in the Efficient-Head module. Through experimental research, comprehensively considering detection accuracy, number of model parameters, computational complexity, and other factors, the optimal value of $g$ was finally determined and set to 16 .

$$k \times k \times c_{\text{in}} \times c_{\text{out}} = k^2 c_{\text{in}} c_{\text{out}} \tag{10}$$

$$h_{\text{in}} \times w_{\text{in}} \times c_{\text{in}} \times c_{\text{out}} \times k \times k = k^2 h_{\text{in}} w_{\text{in}} c_{\text{in}} c_{\text{out}} \tag{11}$$

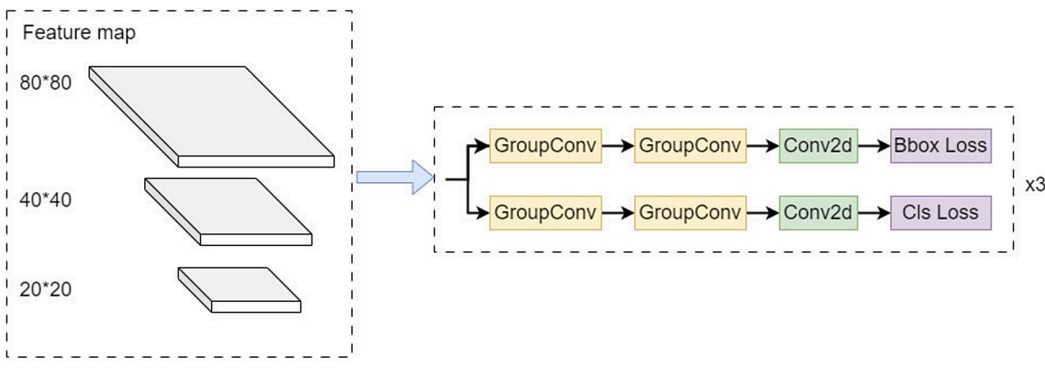

**Fig 13**. **Efficient-Head module structure diagram.**

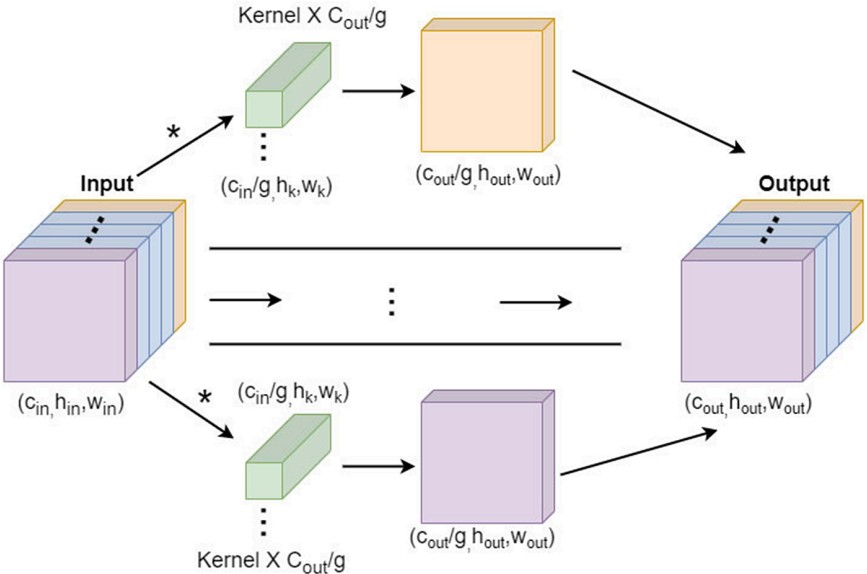

**Fig 14. Group Convolution schematic diagram.**

$$\left(\left(\frac{c_{in}}{g}\right) \times \left(\frac{c_{out}}{g}\right) \times k \times k\right) \times g = \frac{k^2 c_{in}\, c_{out}}{g} \tag{12}$$

$$h_{in} \times w_{in} \times \left(\frac{c_{in}}{g}\right) \times \left(\frac{c_{out}}{g}\right) \times k \times k \times g = \frac{k^2 h_{in}\, w_{in}\, c_{in}\, c_{out}}{g} \tag{13}$$

**(5) Improved YOLO11 model**

In UAV-based tea leaf disease detection tasks, to enhance the real-time performance and precision of the YOLO11 model in complex scenarios, this paper makes optimizations to it. First, for the backbone network, the traditional C3K2 module is replaced with the novel lightweight FC_C3K2 module proposed in this paper. This module dynamically adjusts weights for different input features while maintaining lightweight properties, enhancing the model's adaptability when processing complex or variable inputs. This design not only enhances the flexibility of the model, but also ensures efficient performance in various tasks and environments. Second, for the neck network, the original neck network is replaced with the High-level Screening Feature Pyramid Network (HSFPN). HSFPN enhances the feature expression ability and robustness of the model through the fusion of the features of high and low levels, while effectively reduces the overall size of the model, which makes the model more robust in complex scenarios. Finally, for the detection head part, the Group Convolution (GConv) is adopted instead of traditional convolution, and an Efficient-Head structure is designed. This structure provides a more efficient solution for real-time detection tasks by effectively addressing the overfitting problem of the model, while reducing the training parameters and computational complexity. This paper refers to the YOLO11 model incorporating the FC_C3K2 module, HSFPN module, and Efficient-Head module as YOLO11-FC_C3K2-HSFPN-Efficient-Head (abbreviated as FCHE-YOLO), with its model structure shown in Fig 15.

## Method summary and process description

To enable readers to more clearly understand the various methods, their functions, and interrelationships, this paper systematically integrates the UAV low-altitude remote sensing data acquisition method, the data augmentation method, and

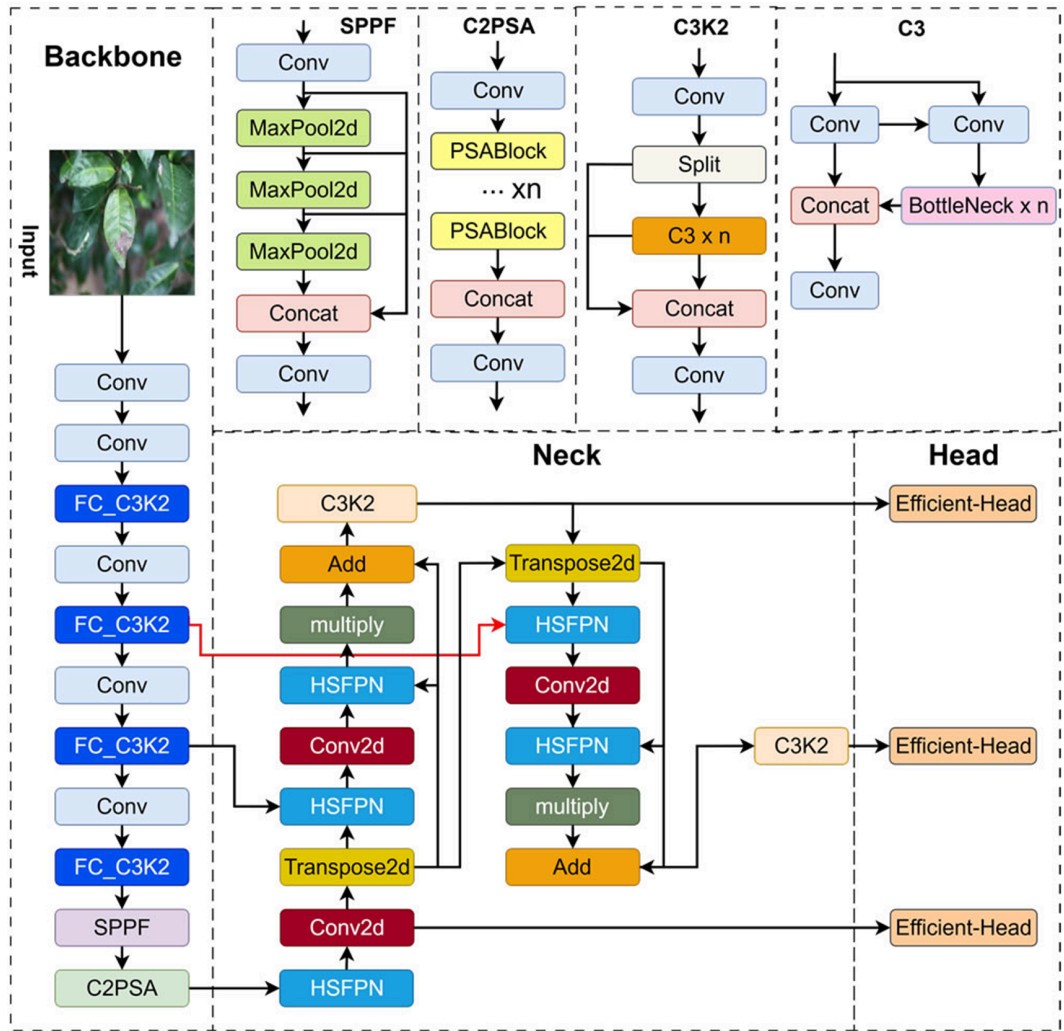

**Fig 15**. **FCHE-YOLO model structure diagram.**

three core modules of the improved YOLO11 model (i.e. FC_C3K2, HSFPN, Efficient-Head). The functions, purposes of each method, and their correspondence with the research objectives are shown in Table 3.

## Experiments and results analysis

### Tea leaf disease dataset

The tea leaf disease images captured by UAVs as described above are annotated using the labelimg annotation software and saved as txt files, adopting the YOLO format [35]. By reading the YOLO annotation files, target category and location information can be obtained for subsequent network training.

Due to the limited scale of the collected dataset and insufficient data volume, the model's ability to detect tea leaf diseases is restricted. To address this issue, this paper expands the dataset through data augmentation techniques to enhance the model's generalization capability. During the data augmentation process, the brightness adjustment amplitude is randomly selected within the range of [-20%, +20%], and the rotation angle is randomly generated within the range of [0°, 360°]. In addtion, we aslo apply random cropping and flipping to increase the diversity of the image. Embedding

**Table 3**. Performance comparison of optimization module ablation experiments.

| Method name | Stage | Main function and role | Problems solved | Relationship with research objectives |
|---|---|---|---|---|
| UAV low-altitude remote sensing image acquisition | Data acquisition stage | Achieve data collection covering whole tea plantation areas via high-resolution UAV aerial photography | Low efficiency and insufficient coverage of traditional manual collection | Provide comprehensive, high-quality image data foundation for model training |
| Data augmentation (brightness adjusting, rotation, cropping, mosaicing) | Data preprocessing stage | Expand sample diversity and improve model generalization ability | Limited data volume and susceptibility to overfitting | Enhance model robustness and stability |
| FC_C3K2 module (FasterNet + CGLU) | Backbone | Extract lightweight feature, and improve feature expression capability in complex backgrounds | Complex backgrounds and difficult feature separation | Enhance the model's ability to extract tea leaf disease features |
| HSFPN module | Neck | High-level screening and multi-scale feature fusion | Poor performance in small target detection | Improve detection accuracy and robustness for diseases at different scales |
| Efficient-Head structure | Head | Use GConv to reduce parameter count and computational complexity | Slow inference speed and overfitting | Improve real-time detection performance and edge deployment efficiency |
| Model evaluation and comparative experiment | Performance verification stage | Validate the improvements by comparing with mainstream models | Lack of systematic verification | Demonstrate the comprehensive advantages of the proposed model in accuracy, speed, and lightweight design |
| Edge deployment | Model deployment stage | Deploy on Jetson TX2 platform and calculate model inference efficiency and power consumption | Limited real-time detection performance | Verify the practical deployment feasibility of the model's lightweight design |

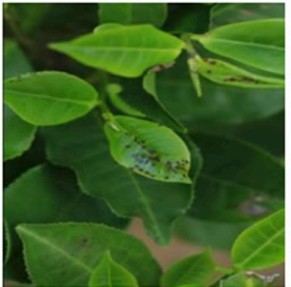 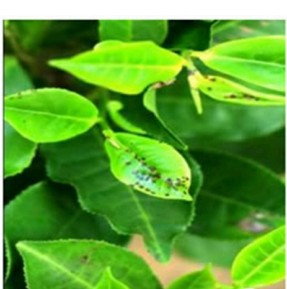 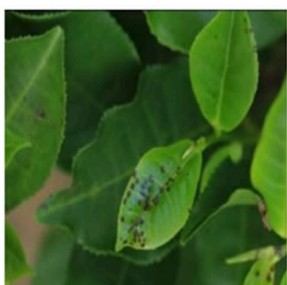 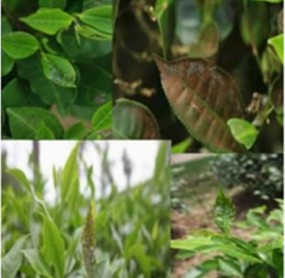

(a) Original image (b) Brightness enhancement (c) Cropping and flipping (d) Mosaic

**Fig 16**. Dataset visualization.

technique is adopted to generate richer samples. Using dataset partitioning code, these 5,400+ images are divided in an 8:1:1 ratio, with 4,328 used for training, 541 for validation, and 540 for testing.

Using visualization tools, the established dataset was visualized as shown in Fig 17, where (a) shows the target box position distribution diagram with coordinates representing center positions, and (b) shows the target box size distribution diagram with coordinates representing annotated width and height.

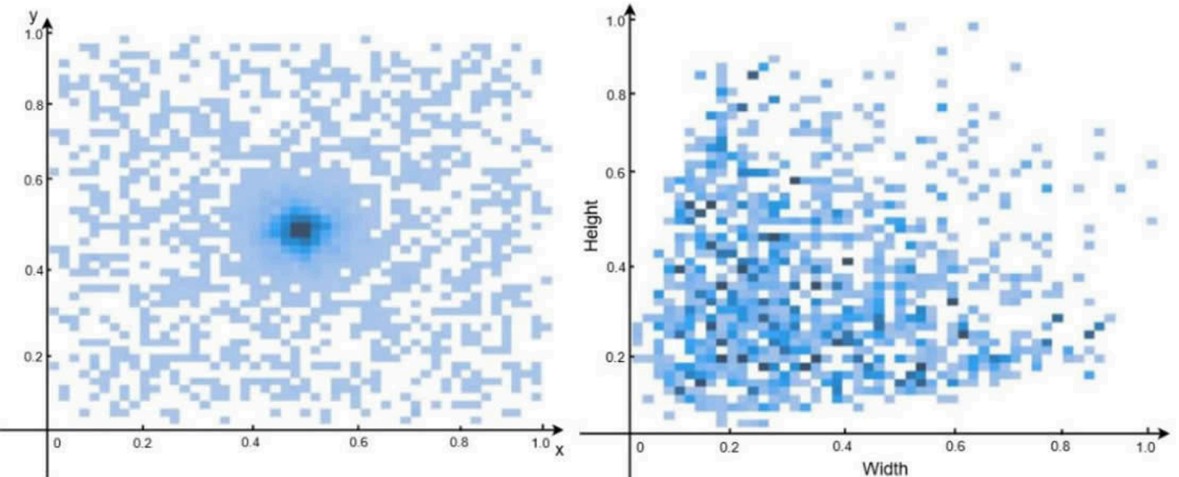

(a) Target bounding box location distribution map (b) Target bounding box size distribution map

**Fig 17**. **Images after data augmentation processing.**

**Table 4**. **Hyperparameter settings.**

| Hyperparameter | bf Value |
|---|---|
| Learning rate | 0.001 |
| Momentum | 0.937 |
| Epochs | 200 |
| Batch size | 24 |
| Image size | $640 \times 640$ |
| Optimizer | SGD |
| Weight decay | 0.0005 |
| Warmup epochs | 3 |
| Warmup momentum | 0.8 |
| Warmup bias learning rate | 0.1 |
| Random seed | 42 |
| Division ratio of dataset | 8:1:1 |

From Fig 16(a), it can be observed that the center positions of target boxes are relatively evenly distributed in the images, indicating that targets are widely and evenly distributed in space. From Fig 16(b), it can be seen that the sizes of the target boxes present a relatively consistent distribution, meaning that most targets do not differ greatly in size.

## Experimental environment

Experiments were carried out on a high-performance computing system with Intel® Core™ i9-9900K CPU and GeForce RTX 2080 Ti 11G GPU. The operating system is Windows 10. PyTorch 1.13.1 is adopted as the deep learning framework, with CUDA version 11.6 and cudnn version 9.6.0, which can fully utilize hardware performance for model training and optimization.

To validate the proposed method's superiority in tea leaf disease detection, this paper used a dataset containing 5,409 images for model training. The YOLO11 model was used as the base model, upon which improvements and optimizations were made. All training processes used YOLO11n.pt pre-trained weights to ensure objective and independent performance evaluation of the improved model. Table 4 shows the specific hyperparameter information for training models.

## Evaluation metrics

To comprehensively evaluate the model's performance on tea leaf disease detection tasks, this paper adopted commonly used evaluation metrics in the field of object detection, including Precision (P), Recall (R), F1 Score (F1), mAP@50 (mean Average Precision), and Frames Per Second (FPS). Additionally, to better reflect the degree of model lightweighting, this paper also adopted metrics such as parameter count (parameters), floating point operations (GFLOPs), and model memory usage.

Precision (P) measures the proportion of true positive samples among those predicted as positive by the model. Recall (R) measures the proportion of actual positive samples that are correctly predicted as positive by the model. F1 Score is the harmonic mean of Precision and Recall, which is used to comprehensively reflect the balance between the model's precision and recall. Mean Average Precision (mAP) is used to calculate the average precision (AP) across multiple categories. It is the process of averaging the precision for each category, as shown in Formulas (14-18):

$$P = \frac{TP}{TP + FP} \tag{14}$$

$$R = \frac{TP}{TP + FN} \tag{15}$$

$$F1 = \frac{2}{\frac{1}{P} + \frac{1}{R}} = \frac{2 * P * R}{P + R} \tag{16}$$

$$mAP = \frac{1}{n} \sum_{k=1}^{n} AP_k \tag{17}$$

$$AP_k = \int_0^1 p(r)dr \tag{18}$$

where $n$ represents the number of categories, and $AP_k$ represents the accuracy of the $k$-th category. True Positive (TP) indicates that the prediction is positive, the actual label is positive, and the prediction is correct; False Negative (FN) indicates that the prediction is negative, the actual label is positive, and the prediction is incorrect; False Positive (FP) indicates that the prediction is positive, the actual label is negative, and the prediction is incorrect; True Negative (TN) indicates that the prediction is negative, the actual label is negative, and the prediction is correct.

## Experiments and analysis

**Ablation experiments.** To evaluate the effect of different modules on model performance, under the experimental conditions as described above, this paper examined the impact of adding each module on P, R, F1 Score, mAP, number of parameters, FLOPs, memory size, and detection rate. Seven groups of ablation experiments were set up for comparison, with results shown in Table 5. Here, "✓" represents strategies used in the improved model, while "×" represents strategies not used. The difference between YOLO11-F and YOLO11-FC is that YOLO11-FC integrates both FasterNet and CGLU in the C3K2 module, while YOLO11-F only integrates the FasterNet module.

The quantitative analysis of the ablation experiment results shows that each improved module's performance has been significantly enhanced. The analysis is as follows:

FasterNet module performance analysis (YOLO11-F): Integrating FasterNet into the C3K2 module improves the performance of the YOLO11 model effectively. The accuracy and the recall rate increase from 95.3% to 95.8% and from 93.4% to 94.1%, respectively, with an absolute increase of 0.5 and 0.7 percentage points and a relative increase rate of 0.52% and 0.75%, respectively. The F1 score also increased from 94.7% to 94.9%, indicating that the FasterNet module plays a positive role in improving the overall detection balance of the model. More importantly, the model lightweighting

**Table 5**. Performance comparison of optimization module ablation experiments.

| Model | FasterNet | FC_ C3K2 | HSFPN | Efficient-Head | P/% | R/% | F1% | mAP 0.5/% | FPS (f/s) | FLOPs /G | Parameters /M | Size (MB) |
|---|---|---|---|---|---|---|---|---|---|---|---|---|
| YOLO11 | × | × | × | × | 95.3 | 93.4 | 94.7 | 94.1 | 43.3 | 6.4 | 2.59 | 5.24 |
| YOLO11-F | ✓ | × | × | × | 95.8 | 94.1 | 94.9 | 94.6 | 44.2 | 5.9 | 2.30 | 4.67 |
| YOLO11-FC | × | ✓ | × | × | 96.5 | 94.6 | 95.1 | 95.3 | 43.5 | 5.7 | 2.24 | 4.57 |
| YOLO11-H | × | × | ✓ | × | 97.5 | 95.1 | 96.4 | 96.2 | 44.8 | 5.7 | 1.83 | 3.76 |
| YOLO11-E | × | × | × | ✓ | 96.2 | 94.5 | 94.1 | 95.1 | 43.5 | 5.2 | 2.32 | 4.70 |
| YOLO11-FCH | × | ✓ | ✓ | × | 97.9 | 97.2 | 96.9 | 97.5 | 44.9 | 5.3 | 1.66 | 3.44 |
| FCCHE-YOLO (Ours) | × | ✓ | ✓ | ✓ | 98.9 | 97.8 | 97.7 | 98.1 | 47.5 | 4.2 | 1.46 | 3.04 |

is achieved with the model performance improved: The FLOPs reduce from 6.4 G to 5.9 G, and the computational complexity is reduced by 7.8%; the parameter count decreases from 2.59 M to 2.30 M, and the compression rate reaches 11.2%; the model size is compressed from 5.24 MB to 4.67 MB, saving 10.9% on storage space. These indicate that the FasterNet module effectively optimizes resource utilization efficiency while maintaining detection accuracy.

Effect analysis of Convolutional Gated Linear Unit (YOLO11-FC): Introduction the CGLU module on the basis of YOLO11-F enhances the model's feature expression ability significantly. CGLU achieves an improvement in local feature capture capability by deeply fusing channel attention mechanism with local image features. The results show that the accuracy rate, the recall rate, and the average accuracy mAP increase from 95.8% to 96.5%, from 94.1% to 94.6%, and from 94.6% to 95.3%, respectively, with an increase of 0.7, 0.5, and 0.7 percentage points (a relative increase of 0.73%, 0.53% and 0.74%), respectively; and the F1 Score increases from 94.9% to 95.1%, further demonstrating that the model has achieved a better balance in accuracy rate and recall rate. At the same time, the model complexity is further optimized: the FLOPs decreases from 5.9 G to 5.7 G (a reduction of 3.4%), the parameter count decreases from 2.30 M to 2.24 M (a reduction of 2.6%), and the model size is compressed from 4.67 MB to 4.57 MB (a compression of 2.1%). The experimental results demonstrate that the CGLU module can improve detection accuracy while optimizing computational efficiency, reflecting the effectiveness and practicality of the design.

Effect analysis of Advanced Screening Feature Pyramid Network (YOLO11-H): The replacement of the original neck network with HSFPN brings the most significant single module performance improvement. The accuracy, the recall rate, and the mAP significantly increase from 95.3% to 97.5%, from 93.4% to 95.1%, and from 94.1% to 96.2%, respectively, with an increase of 2.2, 1.7, and 2.1 percentage points (a relative increase of 2.31%, 1.82% and 2.23%), respectively; and the F1 Score increases from 94.7% to 96.4%, indicating an obvious improvement in the balance between accuracy rate and recall rate of the model. While optimizing multi-scale feature fusion, the model complexity is significantly reduced: the FLOPs are reduced from 6.4 G to 5.7 G, and the computational load is reduced by 10.9%; the parameter count is significantly reduced to 1.83 M (a decrease of 29.3%); the model size is compressed to 3.76 MB (a reduction of 28.2%). HSFPN achieves dual optimization of detection performance and computational efficiency by efficiently screening feature maps and synergistically fusing information at different scales, particularly suitable for handling multi-scale target problems in tea disease detection.

Efficient Head effect analysis (YOLO11-E): The Efficient Head structure uses group convolution instead of traditional convolution operations to improve the efficiency of the detection head. Performance indicators show that the accuracy increases from 95.3% to 96.2%, with an increase of 0.9 percentage points (a relative increase of 0.94%); the recall rate increases from 93.4% to 94.5%, with an increase of 1.1 percentage points (a relative increase of 1.18%); the F1 Score decreases from 94.7% to 94.1%, with a relatively small change, indicating that the module's main attribution lies in optimizing computational efficiency while maintaining balanced performance; the mAP increases from 94.1% to 95.1%, with an increase of 1.0 percentage point (a relative increase of 1.06%). More significantly, the computational

efficiency acquires a significant improvement: the FLOPs decrease from 6.4 G to 5.2 G, and the computational complexity is decreased by 18.8%; the parameter count decreases from 2.59 M to 2.32 M (a decrease of 10.4%); the model size is compressed from 5.24 MB to 4.70 MB (compressed by 10.3%). The results indicate that the group convolution effectively reduces computational cost while maintaining detection accuracy, providing strong support for edge device deployment.

Multi-module collaborative effect analysis (YOLO11-FCH): The collaborative fusion of FC_C3K2 and HSFPN modules has obtained significant performance synergy effect. The accuracy significantly increases from 95.3% to 97.9%, with an increase of 2.6 percentage points (a relative increase of 2.73%); the recall rate increases from 93.4% to 97.2%, with an increase of 3.8 percentage points (a relative increase of 4.07%); the F1 Score increases from 94.7% to 96.9%, demonstrating a significant synergistic gain of multi module fusion in detection stability and balance; the MAP increases from 94.1% to 97.5%, with an increase of 3.4 percentage points (a relative increase of 3.61%). The lightweighting effect of the model is also significant: the parameter count is compressed to 1.66 M (a reduction of 35.9%), and the model size is reduced to 3.44 MB (a reduction of 34.4%). Multi-module collaboration not only achieves the superposition of individual module effect, but also generates additional performance synergy gain, verifying the rationality of the overall architecture design.

Final model comprehensive performance analysis (FCHE-YOLO): The FCHE-YOLO model, which integrates all improved modules, achieves an optimal comprehensive performance. Compared with the benchmark YOLO11, its accuracy increases from 95.3% to 98.9%, with an absolute increase of 3.6 percentage points, and a relative improvement rate of 3.78%; the recall rate increases from 93.4% to 97.8%, with an absolute increase of 4.4 percentage points and a relative increase rate of 4.71%; the F1 Score increases significantly from 94.7% to 97.7%, with an absolute increase of 3.0 percentage points, and a relative improvement rate of 3.17%, indicating that the model has achieved the best balance between high accuracy and high recall; the MAP increases from 94.1% to 98.1%, with an absolute increase of 4.0 percentage points and a relative increase rate of 4.25%; the inference speed increases from 43.3 FPS to 47.5 FPS, with an increase of 9.7%, meeting the real-time detection requirement. The model lightweighting has a significant improvement: The parameter count is greatly reduced from 2.59 M to 1.46 M, with a compression rate of up to 43.6%; the FLOPs decreases from 6.4 G to 4.2 G, reducing computational complexity by 34.4%; the model size is compressed from 5.24 MB to 3.04 MB, saving 42.0% on storage space.

The systematic ablation experiment verification shows that the FCHE-YOLO model successfully integrates four core improved modules: FasterNet, CGLU, HSFPN, and Efficient Head, achieving dual optimization of detection performance and computational efficiency. Compared with the benchmark YOLO11 model, FCHE-YOLO improves the accuracy, recall, F1 Score, and mAP by 3.6%, 4.4%, 3.0% and 4.0%, respectively. Meanwhile, the inference speed is increased by 9.7%, the parameter count is compressed by 43.6%, and the computational complexity is reduced by 34.4%. Each improved module not only plays a significant role independently, but also produces a good synergistic gain effect, which is more important. Ultimately a high-precision (mAP=98.1%), high-efficiency (47.5 FPS), and lightweight (3.04 MB) tea disease detection model has been constructed, providing a practical technical solution for agricultural intelligent detection, with important theoretical value and practical application significance.

In addition, in order to visually demonstrate the model's performance changes before and after improvement during the training process, the change curves of four key indicators i.e. localization loss (Box-loss), classification loss (Cls_loss), average accuracy (mAP_0.5), and comprehensive accuracy (mAP_0.5:0.95), are shown in Fig 18. Among them, Box-loss measures the deviation between the predicted and real boxes, and the smaller the value, the more accurate the positioning; Cls_loss reflects the classification accuracy, with smaller values indicating more precise recognition; MAP_0.5 represents the average accuracy of various types when the IoU threshold is 0.5, and mAP_0.5:0.95 reflects the model performance more comprehensively by further integrating the accuracy performance under multiple thresholds.

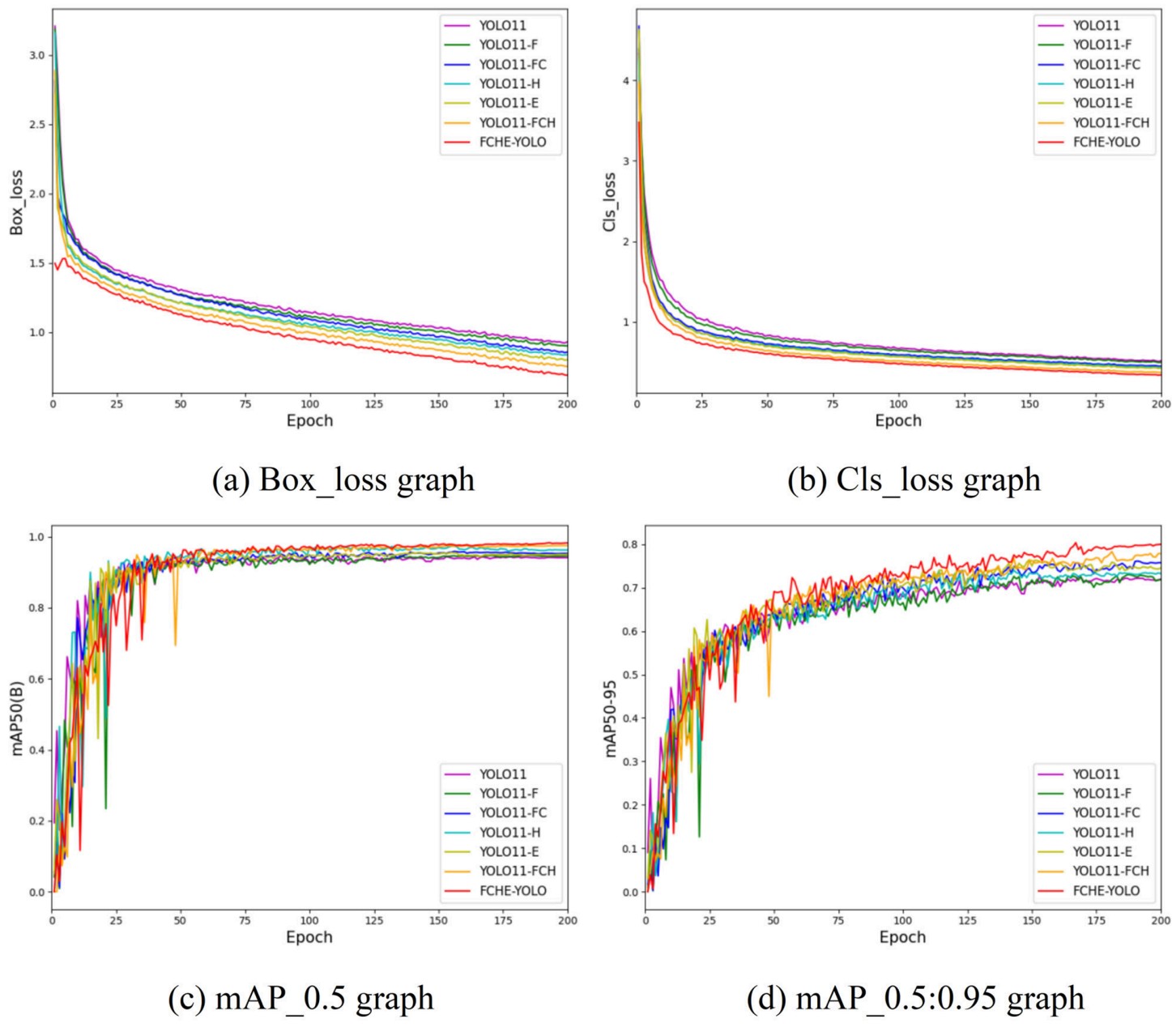

**Fig 18**. Comparison graphs of different metrics.

Fig 18(a) shows that the Box loss curve of the improved FCHE-YOLO model converges faster and achieves lower values compared to other models, indicating improved precision in predicting target positions. Fig 18(b) shows small differences between the models, but the improved FCHE-YOLO model achieves the lowest value, indicating some improvement in target category prediction. Fig 18(c) and 18(d) show the accuracy variation trend of the model at different iteration stages. In the initial stage (Round 0-100), the mAP of each model increases rapidly, and in the later stage (Round 100-200), it tends to stabilize, and the FCHE-YOLO model achieves the highest mAP 0.5 and mAP 0.5:0.95 values, which

are 98.1% and 96.4%, respectively, significantly better than other models. This indicates that the improved FCHE-YOLO model has no adverse effect on convergence speed while effectively improving the accuracy of tea leaf disease detection.

**Comparison of algorithm performance.** To verify the effectiveness of the approach proposed in this paper, several mainstream object detection models were selected for comparative experiments, including Faster RCNN, SSD, YOLO series (v5n, v8n, v10n, v11n), RT-Detr series, and the FCHE-YOLO model proposed in this paper. Table 5 presents the comparison results of different object detection models.

To verify the detection performance of the proposed FCHE-YOLO model, several mainstream object detection models are selected for comparative experiments, which cover different types of network architectures, including YOLO series (v5n, v8n, v10n, v11n), Faster R-CNN, SSD, Transformer structure RT-DETR series models, as well as recently proposed improved versions such as YOLO Tea [21] and YOLOv8-RCAA [24]. The reasons for choosing these models are multifaceted.

Firstly, these models have a wide range of applications in the field of computer vision and perform excellently in their respective domains. YOLO series and Faster R-CNN models are classic models in the object detection field, and their excellent performance is proven in multiple practical tasks. Therefore, choosing these models as comparison objects is to fully verify the advantages of the FCHE-YOLO model. Secondly, the comparison with these models occupying different architecture and characteristics benefits to demonstrate the advantages of FCHE-YOLO from multiple dimensions. The YOLO series models are mainly single-stage detection networks, emphasizing real-time performance and inference speed, especially suitable for edge device applications requiring efficient processing; Faster R-CNN is a two-stage detection network that performs well in accuracy, but has slow inference speed and high computational cost; the RT-DETR model adopts the Transformer architecture and exhibits certain advantages in handling complex target relationships, but large computational load makes it unsuitable for running on resource constrained devices. YOLO-Tea and YOLOv8-RCAA as improved models for tea disease detection are incorporated, providing a comparative perspective to validate the advantages of FCHE-YOLO in this specific task.

The above models covering different types of architectures can verify the performance advantages of FCHE-YOLO by comparison from detection accuracy, model complexity, and deployment efficiency. Table 6 presents the comparative results of different object detection models.

The experimental results show that the improved FCHE-YOLO model performs optimally overall, with mAP 0.5 reaching 98.1%. Compared to Faster R-CNN, SSD, YOLOv5n, YOLOv8n, YOLOv10, RT-Detr, YOLO11, YOLOv8-RCAA and YOLO-Tea models, it shows the mAP is improved by 19.5%, 30.5%, 9.7%, 6.9%, 8.4%, 4.7%, 4%, 0.2% and 5.8% respectively, and the F1 score is increased by 22.5%, 31.3%, 10.9%, 7.1%, 9.6%, 5.4%, 3.8%, 4.7%, and 5.6% respectively. Specifically, Faster R-CNN has a floating-point operation count of 209 G with a large model size and slow detection speed, failing to meet real-time detection requirements. The single-stage detection model SSD, with a weight of 90.1 MB and detection speed of 29.5 frames per second, struggles to meet the real-time and resource-constrained requirements

**Table 6**. **Performance comparison of different detection models.**

| Model | P/% | R/% | F1/% | mAP0.5% | FPS(f/s) | FLOPs/G | Size(MB) |
|---|---|---|---|---|---|---|---|
| Faster RCNN | 79.5 | 78.3 | 73.5 | 78.6 | 6.5 | 209 | 149 |
| SSD | 68.1 | 67.3 | 64.7 | 67.7 | 28 | 29.5 | 90.1 |
| YOLOv5n | 89.5 | 86.1 | 85.1 | 88.4 | 35.6 | 6.3 | 14.7 |
| YOLOv8n | 91.5 | 90.6 | 88.9 | 91.2 | 41.3 | 8.9 | 6.23 |
| YOLOv10 | 89.5 | 90.0 | 86.4 | 89.7 | 39.5 | 6.7 | 5.6 |
| RT-Detr | 93.7 | 93.1 | 90.6 | 93.4 | 37.1 | 21.2 | 63.4 |
| YOLO11 | 95.3 | 93.4 | 92.2 | 94.1 | 43.3 | 6.4 | 5.24 |
| YOLOv8-RCAA | 98.2 | 85.3 | 91.3 | 97.9 | 45.6 | 8.1 | 6.46 |
| YOLO-Tea | - | - | 90.4 | 92.3 | 39.4 | 7.9 | 15.6 |
| FCCHE-YOLO (Ours) | 98.9 | 97.8 | 96.0 | 98.1 | 47.5 | 4.2 | 3.04 |

of UAV devices. Although the RT-Detr model exhibits excellent accuracy, its excessive parameter count makes it unsuitable for real-time detection on edge devices. Compared to the YOLO series algorithms, the improved FCHE-YOLO model performs best in model size, detection speed, and computational complexity, making its overall performance superior. The improved FCHE-YOLO model can meet the requirements for precise detection of tea leaf diseases by UAVs.

In the YOLO series algorithms, YOLOv5n performs well in speed (35.6 FPS), but its accuracy is relatively low, with a mAP of only 88.4%, inferior to FCHE-YOLO. YOLOv8n has a higher accuracy (mAP of 91.2%), a smaller model size (6.23 MB), and a good inference speed (41.3 FPS), yet there is a gap in accuracy and computational complexity compared to FCHE-YOLO. YOLOv10 achieves a good balance between accuracy (mAP of 89.7%) and speed (39.5 FPS), but still falls short of FCHE-YOLO's performance. YOLO11's accuracy and inference speed improve to 94.1% and 43.3 FPS, respectively, close to FCHE-YOLO, however its computational complexity (6.4 G FLOPs) and model size (5.24 MB) are slightly higher. YOLOv8-RCAA has an mAP of 97.9% and an inference speed (45.6 FPS), yet its real-time performance on edge devices is still limited due to its 8.1G FLOPs and large model size of 6.46 MB. The mAP of the YOLO Tea model is 92.3%, but its inference speed (39.4 FPS) is low and model size (15.6 MB) is large, making it insufficient in real-time performance and under resource limitation condition of edge devices. The improved FCHE-YOLO model performs the best in terms of model size, detection speed, and computational complexity respectively, occupying a better overall performance, meeting UAVs' requirements of precise detection of tea leaf diseases.

**Model comparison and innovation analysis.** To further validate the innovation and advantages of FCHE-YOLO compared to existing advanced detection models of tea leaf diseases such as YOLOv8-RCAA, YOLO-Tea, RT-Detr, and ED-Swin Transformer [36], a detailed comparative analysis is conducted. Table 7 shows the differences between FCHE-YOLO and these four models in terms of backbone network structure, feature fusion structure, detection head structure, and inference efficiency.

Table 7 shows the lightweight design and optimization of FCHE-YOLO on multiple key structures: With a self-designed FC_C3K2 module, the backbone network has fewer parameters and lower computational complexity; adopting HSFPN,

Table 7. Comparative analysis of FCHE-YOLO and mainstream lightweight models.

| Model name | Backbone network structure | Feature fusion structure | Detection head structure | mAP_0.5 | Parameter count (M) | FLOPs (G) | Model characteristics and limitations |
|---|---|---|---|---|---|---|---|
| YOLOv8- RCAA | CSPDarknet + RepVGG | PANet + CBAM | Decoupled + ATSS | 97.9 | 5.6 | 5.2 | High accuracy, but low inference speed |
| YOLO-Tea | CSPDarknet53 + GCNet | RFB + ACmix | Common detection head | 92.3 | 15.6 | 7.9 | Good detection accuracy, yet high deployment cost |
| RT-Detr | Transformer Encoder- Decoder | FPN-like structure | Anchor-free Head | 93.4 | 90.3 | 21.2 | Complex large model structure with a poor real-time performance |
| ED-Swin Transformer | EMAGE module | DASPPmodule | Common detection head | 96.5 | 49.3 | 9.3 | Large model occupying strong multi-scale feature processing capability |
| FCHE- YOLO (Ours) | FC C3K2 (lightweight) | HSFPN (attention) | Efficient- Head (GConv) | 98.1 | 1.46 | 4.2 | High detection accuracy, best inference efficiency, suitable for edge deployment |

the feature fusion part enhances the multi-scale feature expression ability combined with the attention mechanism; the detection head adopts an efficient Efficient-Head structure and introduces GConv to improve computational efficiency. FCHE-YOLO achieves the lowest parameter count (1.46 M) and lower FLOPs (4.2 G) while maintaining the highest detection accuracy (98.1%), significantly better than similar models, especially in resource-limited edge devices such as drones.

RT-DETR and ED-Swin Transformer perform well in terms of accuracy, but large parameter couount and low inference speed make them difficult to be effectively deployed in UAV devices. Considering the practical application scenarios i.e. UAVs and embedded devices, FCHE-YOLO via its efficient design can run better in resource-constrained environment while meeting detection accuracy requirement.

In summary, FCHE-YOLO balances detection accuracy, computational efficiency, and model lightweight in structural design, particularly suitable for deployment in edge devices such as UAVs and embedded systems, demonstrating better application potential.

## Radar chart comparison

To intuitively display the comprehensive performance of different models, this paper compares the models from Table 5 across multiple evaluation metrics, including precision (P), recall (R), mAP 0.5, average inference time, parameter count, and model size, visualized through a radar chart [36] as shown in Fig 19. In the radar chart, each curve represents a model, with points closer to the edge indicating better performance on the corresponding metric; a larger enclosed area represents a stronger comprehensive performance.

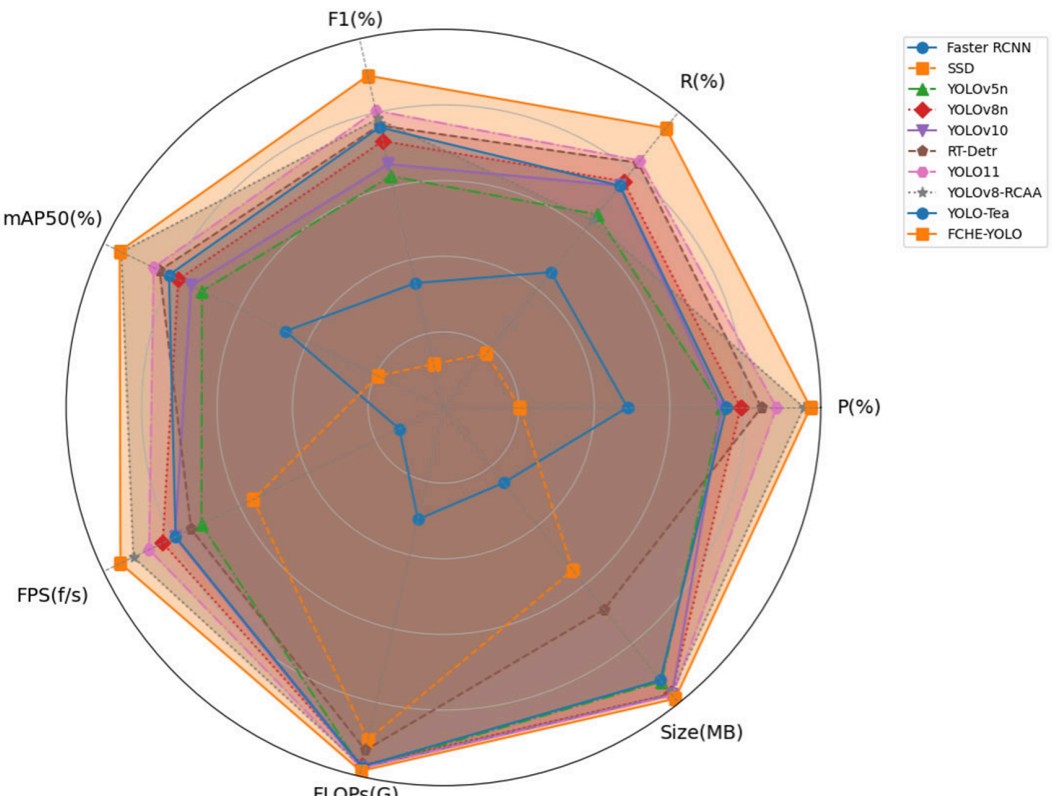

**Fig 19**. **Multi-model performance comparison radar chart.**

From Fig 18, although some models (such as RT-Detr series and YOLOv8 series) achieve good results in mAP_0.5, the FCHE-YOLO model demonstrates more prominent comprehensive performance, especially with clear advantages in the number of model parameters and detection speed. These advantages are crucial for model deployment and real-time requirements in practical applications, giving FCHE-YOLO an edge in practical scenarios.

From Fig 19, it can be seen that FCHE-YOLO exhibits a relatively balanced performance across all dimensions. Especially in the precision (mAP_0.5), F1 score (F1), precision (P), and recall (R), its curves are close to the edge of the radar chart, showing an excellent performance in detection accuracy and category discrimination. FCHE-YOLO can identify the targets of different categories accurately and effectively, and reduce missed and false detections.

In computational complexity and model volume, the FCHE-YOLO curve is closest to the center of the graph, occupying a lowest computational cost and a smallest model volume. These features make it very suitable for deployment in resource-constrained environment, especially when conducting real-time detection on edge devices such as drones, it can effectively reduce computational burden and improve inference speed.

As for RT-DETR, its accuracy is close to FCHE-YOLO and its performance is also good in mAP and F1 scores, but it has a large parameter count and a high FLOPs of up to 21.2 G, far higher than FCHE-YOLO (4.2 G), unsuitable for deployment on edge devices with limited computing power. YOLOv5n and YOLO-Tea have smaller model sizes and lower computational costs, but their detection accuracy and recall are significantly lower than FCHE-YOLO, especially in complex backgrounds false or missed detection are prone to occur.

Through comparison, the radar chart further confirms the comprehensive advantages of FCHE-YOLO in accuracy, real-time performance, and deployment efficiency, which are crucial for model deployment, real-time requirement, and adaptability to resource-constrained devices in practical applications. Therefore, FCHE-YOLO, occupying more advantageous in tea leaf disease detection, can provide efficient and high-precision detection results, especially on edge devices such as UAVs.

## Detection result comparison

To better demonstrate the detection effects of the improved network, this paper selected some UAV aerial test set images and compared the detection results using YOLO11 and the improved FCHE-YOLO model, as shown in Figs 20 and 21.

By comparing Figs 20(a), 20(b) and 21(a), 21(b), it can be seen that the FCHE-YOLO model has a higher detection confidence than YOLO11. The comparison between Figs 20(c) and 21(c) reveals that the FCHE-YOLO model does not miss detections, while YOLO11 exhibits missed detection phenomena. Comparing Figs 20(d) and 21(d), it can be seen

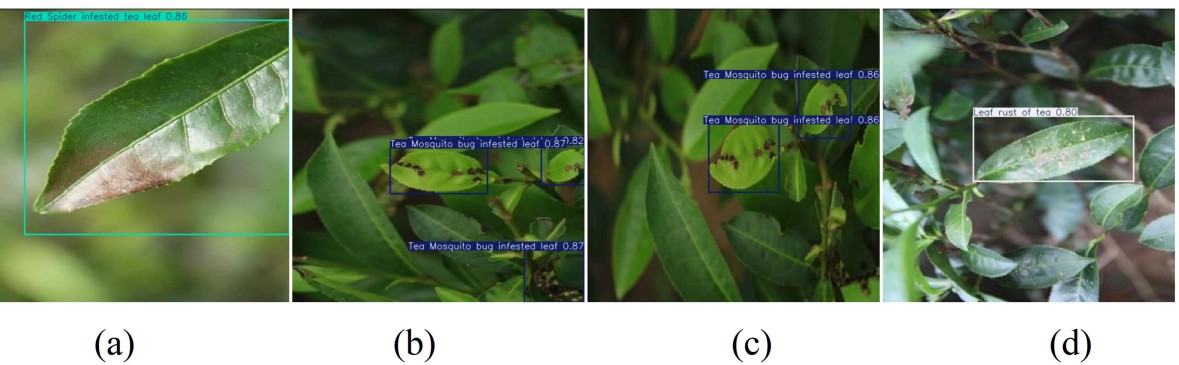

(a)　　　　　　　　　(b)　　　　　　　　　(c)　　　　　　　　　(d)

**Fig 20. YOLO11 detection result images.**

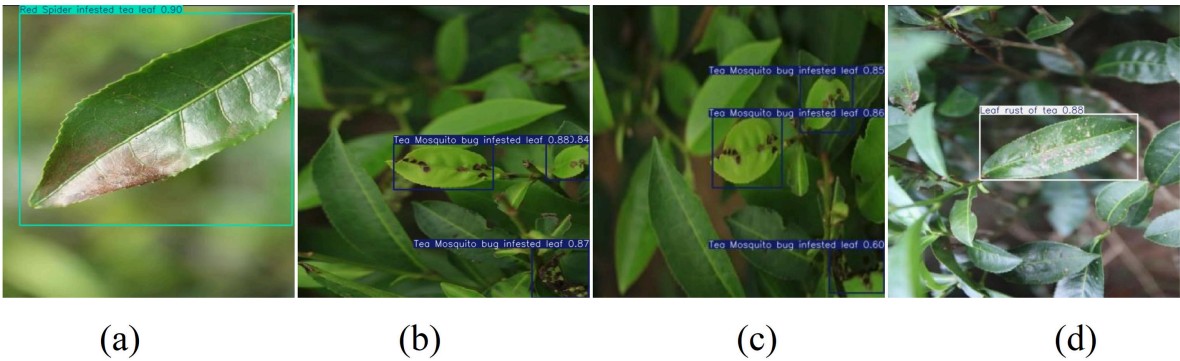

<div align="center">(a) (b) (c) (d)</div>

**Fig 21**. **FCHE-YOLO detection result images.**

that the FCHE-YOLO model has a higher detection confidence than YOLO11. In summary, the FCHE-YOLO model effectively mitigates issues such as low confidence and missed detections that exist in YOLO11 during the detection process, effectively improving detection accuracy.

To further intuitively demonstrate the detection effects of different models, heatmap processing was applied to the original image, with results shown in Fig 22. This figure displays the heatmap results of the Faster RCNN, SSD, YOLOv5n, YOLOv8n, YOLOv10n, RT-Detr, YOLO11, YOLOv8-RCAA, YOLO-Tea, and FCHE-YOLO models, and intuitively presents the possibility of model target detection through color temperature changes.

It can be seen that the heat maps of Faster RCNN, SSD, YOLOv5n, YOLOv8n, and YOLOv10n models are relatively scattered, in which environment and targets are difficult to be distinguished effectively, and targets are not focused accurately. Benefiting from the Transformer based architecture, RT-Detr has a strong focusing ability, which yet leads to the omission of small objects around the target. YOLO11 is difficult to distinguish between targets and backgrounds and identify target locations accurately. Especially in complex backgrounds, false or missed detection is prone to occur, easily

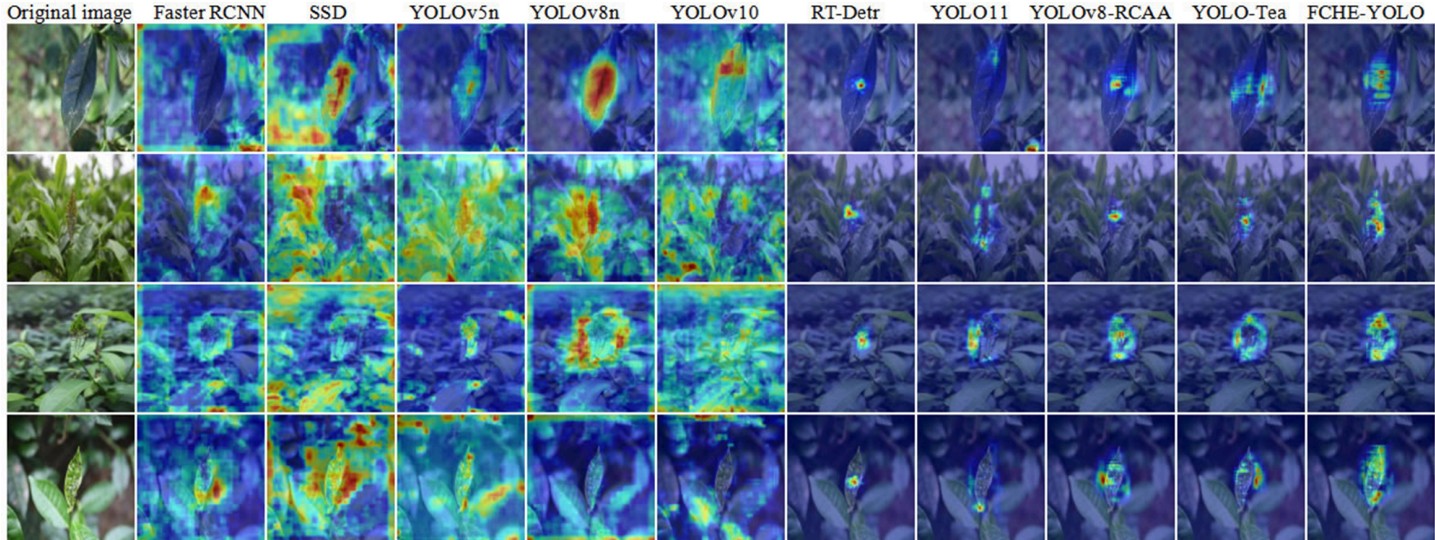

**Fig 22**. **Heatmap comparison figure.**

misidentifying similar background objects as the target or failing to detect real target. The heatmaps of YOLOv8-RCAA and YOLO-Tea have a certain sensitivity to the target, but missed detection still exists. In contrast, FCHE-YOLO is more sensitive to targets and less affected by environmental interference. Although it cannot completely avoid background interference, FCHE-YOLO can more accurately identify and focus on target objects in complex background situations, and significantly enhances the ability to recognize targets in similar backgrounds.

## Conclusions and outlook

### Conclusions

Tea leaf diseases have a significant effect on the yield and quality of tea. High-precision automatic detection and identification of tea leaf diseases are beneficial for precise disease prevention and control. This study proposes an improved YOLO11 model, called FCHE-YOLO, to improve the accuracy and real-time capability of tea leaf disease detection, and embeds it into UAV devices to meet practical application demands.

To address the low accuracy, excessive computational load, and overfitting of current detection methods in complex backgrounds, multiple optimizations are conducted based on YOLO11:

(1) FC_C3K2 module: In the backbone network, a new lightweight module FC_C3K2 is introduced, which combines FasterNet Block, Convolutional Gated Linear Unit (CGLU), and C3K2 structure. This module can significantly reduce the parameter count and computational complexity, and enhance the robustness of the model in complex background, especially with good adaptability on edge devices. This optimization addresses the accuracy shortcomings of existing technologies when handling complex backgrounds.

(2) HSFPN module: In the neck network, the High-level Screening Feature Pyramid Network (HSFPN) is adopted to replace the traditional feature pyramid network. HSFPN enhances the multi-scale detection capability and robustness of the model by efficiently integrating features of different levels, while effectively reducing the model's storage and computational load. This optimization not only enhances the model accuracy, but also makes the improved YOLO11 lighter, more efficient, and particularly suitable for edge device deployment.

(3) Efficient-Head module: In the detection head section, an Efficient-Head structure is designed. It uses Group Convolution (GConv) instead of traditional convolution, thereby reducing the training parameters and computational complexity of the model. This module effectively suppresses overfitting of the model while improving detection accuracy, ensuring good performance on resource-constrained devices.

The results based on the self built dataset from UAVs show that FCHE-YOLO improves the average recognition accuracy (mAP) by 4%, and recognition speed (FPS) by 9%, and reduces the floating-point operations (FLOPs) by 34.3%, parameter count by 38.9%, and memory size by 41.9% compared to the YOLO11 model. Hence, the FCHE-YOLO model not only has a higher accuracy and a faster speed, but also effectively reduces the missed detection in complex backgrounds.

Through these innovative optimizations, FCHE-YOLO demonstrates its advantages in edge computing devices (such as UAVs), delivering an efficient and deployable solution for tea leaf disease detection. These findings provide important references for the future application of object detection methods based on deep learning in agricultural monitoring.

### Outlook

At present, the detection speed of FCHE-YOLO model on desktop GPU devices can reach 47.5 frames per second, showing a good real-time detection capability. However, when deployed to edge devices such as Jetson TX2, the detection speed drops to 15.1 frames per second, equivalent to an average inference delay of 66 ms. Due to the limited computing resources of edge devices, real-time response capability faces certain challenges. To further analyze the feasibility of actual deployment, this paper further tests the performance of FCHE-YOLO on Jetson TX2, and the relevant indicators are shown in Table 8.

**Table 8**. Operating performance indicators of FCHE-YOLO on the Jetson TX2 platform.

| Indicator | Value | Description |
| --- | --- | --- |
| Inference delay | About 66 ms approximately 66 ms | Average inference time per image |
| Detecting frame rate | 15.1 FPS | Tested frame rate, supporting low-speed real-time detection |
| Power consumption | About 4.3 W | Tested average power consumption (during operation) |
| Video memory usage | About 1.2 GB | GPU memory space required for stable operation |

When the UAV equipped with Jetson TX2 performs field detection and spraying operations in tea plantations, the overall power consumption of the system varies with the operation status: in the no-load state (without operation load), the power consumption is approximately 450–500 W, the energy consumption per operation is about 0.20 kWh, and the theoretical endurance is around 1.7 hours (102 minutes); in the full-load state (with a 20 kg load for spraying operations), the power consumption is approximately 600–650 W, the energy consumption per operation is about 0.26 kWh, and the theoretical endurance is around 1.3 hours (78 minutes). The above-mentioned power consumption includes that of the flight motor, spraying/seeding pump, front and rear radars, and edge computing module.

Although currently the model has preliminary edge deployment capability, there is still room for further optimization. In the future, it is planned to improve the model and accelerate its detection through model pruning [37], and TensorRT [38] and quantization deployment methods. Additionally, regarding UAV shooting path planning, currently a simple rectangular flight path is employed, which works well in some tea gardens. The later plan is to optimize UAV shooting paths by leveraging the adaptability and efficiency of improved ant colony algorithms [39], which will enhance overall efficiency and ensure precise coverage of all areas that need to be detected. It should be noted that the data used in this paper is mainly sourced from the Siwangshan Tea Plantation in Xinyang, Henan Province. The data has strong regional characteristics, which may lead to limited generalization ability of the model in other regions or on different tea varieties. In addition, the number of currently collected samples is still relatively limited. In the future, the dataset will be further expanded on this basis, and tea plantation data from other regions (such as Anxi, Wuyishan, etc.) will be collected to verify the robustness and generalization ability of the model in different environments.

## Author contributions

**Conceptualization:** Yaojun Zhang.

**Data curation:** Guiling Wu, Yaojun Zhang, Jianbo Shen.

**Formal analysis:** Yaojun Zhang.

**Funding acquisition:** Yaojun Zhang.

**Investigation:** Guiling Wu, Jianbo Shen.

**Methodology:** Guiling Wu.

**Project administration:** Yaojun Zhang.

**Resources:** Jianbo Shen.

**Software:** Yaojun Zhang.

**Supervision:** Guiling Wu, Jianbo Shen.

**Validation:** Guiling Wu, Jianbo Shen.

**Visualization:** Yaojun Zhang, Jianbo Shen.

**Writing – original draft:** Yaojun Zhang.

**Writing – review & editing:** Guiling Wu, Jianbo Shen.

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
