## [Decision Letter · Decision Letter 0]

8 Jul 2025

PONE-D-25-19024Precise Tea Leaf Disease Detection Using UAV Low-Altitude Remote Sensing and Optimized YOLO11 ModelPLOS ONE

Dear Dr. Wu,

Thank you for submitting your manuscript to PLOS ONE. After careful consideration, we feel that it has merit but does not fully meet PLOS ONE’s publication criteria as it currently stands. Therefore, we invite you to submit a revised version of the manuscript that addresses the points raised during the review process.

Comments to the Author:

(There are no comments. Please check to see if comments were included as a file attachment with this e-mail or as an attachment in your Author Center.)

We look forward to receiving your revised manuscript.

Kind regards,

Farman Ullah

Academic Editor

PLOS ONE

Journal Requirements:

The authors extend their appreciation to the Science and Technology Research Projects of Henan Province (Grant No. 252102111173) and Cangnan County Modern Agricultural Industry Enhancement Project (Grant No. 2024CNYJY08).

This research was funded by Science and Technology Research Projects of Henan Province (Grant No. 252102111173) and Cangnan County Modern Agricultural Industry Enhancement Project (Grant No. 2024CNYJY08). The funders had no role in the design of the study; in the collection, analyses, or interpretation of data; in the writing of the manuscript; or in the decision to publish the results.

The authors declare no conflicts of interest.

5. In the online submission form, you indicated that the data used in this study is available upon request.

Additional Editor Comments:

Comments to the Author:

(There are no comments. Please check to see if comments were included as a file attachment with this e-mail or as an attachment in your Author Center.)

Reviewers' comments:

Reviewer's Responses to Questions

**Comments to the Author**

1. Is the manuscript technically sound, and do the data support the conclusions?

Reviewer #1: Yes

Reviewer #2: Partly

2. Has the statistical analysis been performed appropriately and rigorously?

Reviewer #1: Yes

Reviewer #2: No

3. Have the authors made all data underlying the findings in their manuscript fully available?

Reviewer #1: No

Reviewer #2: Yes

4. Is the manuscript presented in an intelligible fashion and written in standard English?

Reviewer #1: Yes

Reviewer #2: Yes

5. Review Comments to the Author

Reviewer #1: The manuscript addresses an important and current challenge in agricultural monitoring achieving accurate and real-time detection of tea leaf diseases through the integration of low-altitude remote sensing via Unmanned Aerial Vehicles (UAVs) and an enhanced, lightweight deep learning framework, namely the FCHE-YOLO model, which is an optimization of the YOLO11 architecture. The study is technically well-founded and represents a meaningful contribution to the ongoing development of UAV-assisted plant health monitoring systems, which are critical for promoting sustainable agricultural practices and mitigating crop losses due to pests and diseases.

The study is technically competent and makes an important contribution to the continuous development of UAV-assisted plant health monitoring systems, which are vital for encouraging sustainable agricultural practices and decreasing crop losses due to pests and diseases. Overall, the paper is clearly written, logically structured, and supported by a robust set of experimental results that demonstrate the potential benefits of the proposed improvements. Nevertheless, in order to meet the publication standards of PLOS ONE and to maximize the impact and reproducibility of the work, several aspects require further elaboration.

Major comments

• The manuscript introduces a novel backbone module (FC_C3K2), a redesigned neck structure (HSFPN), and a lightweight detection head (Efficient-Head), all integrated into the YOLO11 framework. While these modular enhancements are technically reasonable and well-supported by the experiments, the degree of incremental novelty compared to recent lightweight YOLO versions, such as YOLOv8 or other contemporary architectures, is not yet clearly demonstrated.

To strengthen the contribution and improve clarity for readers, I suggest the authors to provide a more rigorous discussion quantifying and explaining how the proposed FCHE-YOLO architecture differs both functionally and conceptually from existing lightweight variants, including examples like YOLOv8-RCAA and YOLO-Tea. Additionally, presenting a comparative table that outlines the module-wise differences would greatly help to highlight the specific innovations introduced in this study.

• The dataset used in this study consists of approximately 5,400 images obtained through data augmentation, all originally collected from a single tea plantation (Siwangshan). This limited source of data raises valid concerns regarding the potential for overfitting and questions the generalizability of the model to other locations and conditions. Moreover, The data augmentation techniques could be described with precise parameter ranges (e.g., how much brightness adjustment? what rotation angles?).

I suggest the authors to provide a thorough discussion of possible site-specific biases and clarified whether data from additional plantations, different seasons, or varying lighting conditions were included in the dataset or are planned for future work.

• The comparative experiments presented in the manuscript primarily concentrate on conventional models such as Faster R-CNN, SSD, and earlier versions of the YOLO series. However, the recent advancements in transformer-based or hybrid CNN-transformer detection frameworks, such as DE-TR and Swin-Transformer, are not addressed or evaluated in this study.

I recommend the authors to discuss why transformer-based or attention-heavy detectors were not compared or tested. Even if not deployed on UAVs due to resource constraints, this would contextualize the chosen architecture.

• Although the model demonstrates high frame rates on a desktop GPU, its performance decreases to approximately 15 FPS when deployed on edge UAV devices, which could limit its real-time effectiveness in the field. While the manuscript briefly mentions plans for model pruning and TensorRT optimization, I recommend authors to include more specific details on current inference latency and energy consumption during UAV operations. Providing this information would offer a clearer understanding of the model’s practical feasibility and highlight any technical constraints that need to be addressed for successful real-world deployment.

• Many hyperparameters are listed, but key training details such as random seed, augmentation pipeline code, and training/validation split reproducibility are missing. I recommend authors to consider adding the full training pipeline and data splits to a supplementary file or public repository to meet PLOS ONE’s data availability standards.

Minor Comments

• The manuscript has minor grammatical issues and redundant phrases. For example, “the model significantly improves the adaptability and expressiveness” could be shortened, in addition, some references are embedded mid-sentence and break the flow

I Recommend authors a careful language editing and to ensure consistent citation style;

• Figures illustrating the system architecture and modules should be labeled more clearly and be of higher resolution. Some figures (heatmaps, detection examples) are low-resolution and need clearer labeling.

I recommend to use high-resolution annotated figures with consistent color maps and scale bars.

• Some cited papers (e.g., [20], YOLO-Tea) should be cross-checked for accuracy and updated if more recent works exist.

Reviewer #2: The review has been attached for the consideration of authors. There are few key areas that require modification and further test results need to be provided. The paper provide an interesting approach and if revisions are performed it can be put for a review again.

6. PLOS authors have the option to publish the peer review history of their article (what does this mean?). If published, this will include your full peer review and any attached files.

Reviewer #1: No

Reviewer #2: No

---

## [Author Response · Author response to Decision Letter 1]

20 Aug 2025

We are grateful for your time and consideration. Thank you for your attention to our submission.

---

## [Decision Letter · Decision Letter 1]

22 Oct 2025

PONE-D-25-19024R1Precise Tea Leaf Disease Detection Using UAV Low-Altitude Remote Sensing and Optimized YOLO11 ModelPLOS ONE

Dear Dr. Wu,

Thank you for submitting your manuscript to PLOS ONE. After careful consideration, we feel that it has merit but does not fully meet PLOS ONE’s publication criteria as it currently stands. Therefore, we invite you to submit a revised version of the manuscript that addresses the points raised during the review process.

We look forward to receiving your revised manuscript.

Kind regards,

Farman Ullah

Academic Editor

PLOS ONE

Journal Requirements:

Reviewers' comments:

Reviewer's Responses to Questions

**Comments to the Author**

1. If the authors have adequately addressed your comments raised in a previous round of review and you feel that this manuscript is now acceptable for publication, you may indicate that here to bypass the “Comments to the Author” section, enter your conflict of interest statement in the “Confidential to Editor” section, and submit your "Accept" recommendation.

Reviewer #1: All comments have been addressed

Reviewer #2: (No Response)

2. Is the manuscript technically sound, and do the data support the conclusions?

Reviewer #1: Yes

Reviewer #2: Yes

3. Has the statistical analysis been performed appropriately and rigorously?

Reviewer #1: Yes

Reviewer #2: Yes

4. Have the authors made all data underlying the findings in their manuscript fully available?

Reviewer #1: Yes

Reviewer #2: (No Response)

5. Is the manuscript presented in an intelligible fashion and written in standard English?

Reviewer #1: Yes

Reviewer #2: No

6. Review Comments to the Author

Reviewer #1: The revised manuscript shows clear improvements in both clarity and scientific rigor. The authors addressed the major concerns raised in the first review, and the paper is now significantly stronger.

Strengths of the Revision:

- The novelty of the FCHE-YOLO framework is now articulated more clearly, supported by a comparative table against other YOLO variants.

- The dataset section has been expanded, with explicit acknowledgment of site-specific limitations and suggestions for future multi-site validation.

- Transformer-based methods are now discussed, and the rationale for focusing on CNN-based architectures is convincingly explained.

- Figures and tables are clearer, and the manuscript benefits from improved language editing.

The remaining points are minor refinements that could further strengthen the paper: (i) briefly reiterate dataset limitations in the Conclusions for balance, (ii) include approximate energy consumption data from UAV deployment to reinforce field feasibility, and (iii) ensure the data and code repository fully complies with PLOS ONE standards, with unrestricted access to raw data, annotations, augmentation scripts, and trained weights.

A final proofreading pass is recommended to eliminate minor redundancies and improve flow

Conclusion:

The manuscript has been substantially strengthened in response to reviewer feedback. The technical soundness is solid, the results are convincing, and the revisions significantly improve clarity and reproducibility. Some minor refinements remain, particularly regarding the explicit availability of data/code and the contextualization of the model’s novelty and deployment feasibility.

Reviewer #2: Please check the attached review file. Please provide response letter against each of the comment and also indicate where your changes are in the paper.

7. PLOS authors have the option to publish the peer review history of their article (what does this mean?). If published, this will include your full peer review and any attached files.

Reviewer #1: No

Reviewer #2: No

---

## [Author Response · Author response to Decision Letter 2]

17 Nov 2025

We sincerely thank Reviewers #1 and #2 for their insightful feedback, confirm all comments—including discussions on dataset limitations, UAV energy consumption data, repository compliance, backend system introduction, figure/format corrections, and F1 score supplementation—have been fully addressed to enhance our manuscript’s clarity, rigor, and adherence to PLOS ONE’s requirements, and verify the required submission materials have been uploaded to the Editorial Manager system for final assessment.

---

## [Decision Letter · Decision Letter 2]

27 Jan 2026

Precise Tea Leaf Disease Detection Using UAV Low-Altitude Remote Sensing and Optimized YOLO11 Model

PONE-D-25-19024R2

Dear Dr. Wu,

We’re pleased to inform you that your manuscript has been judged scientifically suitable for publication and will be formally accepted for publication once it meets all outstanding technical requirements.

Kind regards,

Chong Xu

Academic Editor

PLOS One

Additional Editor Comments (optional):

Reviewers' comments:

Reviewer's Responses to Questions

**Comments to the Author**

1. If the authors have adequately addressed your comments raised in a previous round of review and you feel that this manuscript is now acceptable for publication, you may indicate that here to bypass the “Comments to the Author” section, enter your conflict of interest statement in the “Confidential to Editor” section, and submit your "Accept" recommendation.

Reviewer #1: All comments have been addressed

2. Is the manuscript technically sound, and do the data support the conclusions?

Reviewer #1: Partly

3. Has the statistical analysis been performed appropriately and rigorously?

Reviewer #1: Yes

4. Have the authors made all data underlying the findings in their manuscript fully available?

Reviewer #1: Yes

5. Is the manuscript presented in an intelligible fashion and written in standard English?

Reviewer #1: Yes

6. Review Comments to the Author

Reviewer #1: I would like to thank the authors for their thoughtful and thorough revisions. The manuscript has improved substantially through each round, and the current version (R2) fully addresses the remaining concerns raised in the previous reviews. The study now meets the scientific, technical, and transparency standards expected by PLOS ONE. The FCHE-YOLO model represents a meaningful advancement in lightweight, UAV-based plant disease detection, with potential applications in real-world agricultural monitoring.

7. PLOS authors have the option to publish the peer review history of their article (what does this mean?). If published, this will include your full peer review and any attached files.

Reviewer #1: No

---

## [Editor Report · Acceptance letter]

PONE-D-25-19024R2

PLOS One

Dear Dr. Wu,

I'm pleased to inform you that your manuscript has been deemed suitable for publication in PLOS One. Congratulations! Your manuscript is now being handed over to our production team.

Kind regards,

on behalf of

Dr. Chong Xu

Academic Editor

PLOS One